

# Data assimilation with multiple types of observation boreholes via ensemble Kalman filter embedded within stochastic moment equations

Chuan-An Xia[1,2], Xiaodong Luo[3], Bill X. Hu[1*], Monica Riva[2,4], Alberto Guadagnini[2,4*]

[1]Institute of Groundwater and Earth Science, Jinan University, Guangzhou, China
[2]Dipartimento di Ingegneria Civile e Ambientale, Politecnico di Milano, Milan, Italy
[3]Norwegian Research Centre (NORCE), Bergen, Norway
[4]Department of Hydrology and Atmospheric Sciences, The University of Arizona, Tucson, USA

*Correspondence to*: Bill X. Hu (bill.x.hu@gmail.com);Alberto Guadagnini(alberto.guadagnini@polimi.it)

**Abstract.** We employ an approach based on ensemble Kalman filter coupled with stochastic moment equations (MEs-EnKF) of groundwater flow to explore the dependence of conductivity estimates on the type of available information about hydraulic heads in a three-dimensional randomly heterogeneous field where convergent flow driven by a pumping well takes place. To this end, we consider three types of observation devices, corresponding to (*i*) multi-node monitoring wells equipped with packers (Type A), (*ii*) partially (Type B) and (*iii*) fully (Type C) screened wells. We ground our analysis on a
variety of synthetic test cases associated with various configurations of these observation wells. Moment equations are approximated at second order (in terms of the standard deviation of the natural logarithm, *Y*, of conductivity) and are solved by an efficient transient numerical scheme proposed in this study. The use of an inflation factor imposed to the observation error covariance matrix is also analyzed to assess the extent at which this can strengthen the ability of the MEs-EnKF to yield appropriate conductivity estimates in the presence of a simplified modeling strategy where flux exchanges between
monitoring wells and aquifer are neglected. Our results show that (*i*) the configuration associated with Type A monitoring wells leads to conductivity estimates with the (overall) best quality; (*ii*) conductivity estimates anchored on information from Type B and C wells are of similar quality; (*iii*) inflation of the measurement-error covariance matrix can improve conductivity estimates when an incomplete/simplified flow model is adopted; and (*iv*) when compared with the standard Monte Carlo -based EnKF method, the MEs-EnKF can efficiently and accurately estimate conductivity and head fields.

## 1 Introduction

Parameter estimation for groundwater system modeling is a key and important challenge, due to our incomplete knowledge of the spatial distributions of hydrogeological attributes, such as hydraulic conductivity. The ensemble Kalman filter (EnKF, Evensen, 1994) is a powerful approach to parameter estimation in subsurface flow (Hendricks Franssen and Kinzelbach, 2008; Zheng et al., 2019) and solute transport (Liu et al., 2008; Li et al., 2012; Chen et al., 2018; Xu and Gómez-Hernández, 2018) scenarios. Estimated system parameters can include conductivity/permeability (Zovi et al., 2017; Botto et al., 2018),



porosity (Li et al., 2012), specific storage (Hendricks Franssen et al., 2011), dispersivity (Liu et al., 2008), river bed conductivity (Kurtz et al., 2014), or unsaturated flow characteristic quantities (Zha et al., 2019; Li et al., 2020).

EnKF can assimilate data sequentially through a real-time updating process. Alternatively, all collected measurements can be assimilated simultaneously, for example within a typical model calibration framework. With reference to the latter aspect,
EnKF becomes an ensemble smoother (ES, van Leeuwen and Evensen, 1996), as it is associated with a smoothing probability density function (PDF) rather than a filtering PDF (Jazwinski, 1967). With reference to the ES, observations in the past and current stages are assimilated only once, thus yielding increased efficiency with respect to EnKF (Skjervheim et al., 2011). Iterative forms of EnKF and ES, usually denoted by IEnKF (Gu and Oliver, 2007; Sakov et al., 2012; Gharamti et al., 2015; Luo, 2014) and IES (Chen and Oliver., 2013; Emerick and Reynolds, 2013; Luo et al. 2015; Chang et al., 2017; Li
et al., 2018), have been developed to improve assimilation performance in scenarios characterized by strongly nonlinear behaviors. A variety of studies investigate challenges linked to such (ensemble) data assimilation algorithms, including, e.g., the possibility of coping with non-Gaussian model parameter distributions (Zhou et al., 2011; Li et al., 2018), physical inconsistency/unphysical results stemming from the estimation workflow (Wen and Chen, 2006; Song et al., 2015), or spurious correlations (Panzeri et al., 2013; Bauser et al., 2018; Luo et al., 2018; Soares et al., 2019). All of these works
contribute to improve the robustness of these algorithms for parameter estimation in complex environmental systems.

Recent studies include the work of Xia et al. (2018), who tackle conductivity estimation in a two-dimensional variable-density flow setting using a localized IEnKF to balance central processing unit (CPU) time and estimation accuracy. Bauser et al. (2018) develop an adaptive covariance inflation method for the EnKF to reduce the negative effect of spurious correlations and illustrate an application of the method in a soil hydrology field context. Mo et al. (2019) use a deep-learning
based model as a surrogate of a solute transport model to reduce the CPU time associated with ensemble-based data assimilation through an iterative local update ensemble smoother in a contaminant identification problem considering a synthetic two-dimensional heterogeneous conductivity field. Li et al. (2020) compare benefits and drawbacks of embedding machine-learning-based (artificial neural network, ANN) and physics-based models into an IES for a set of synthetic unsaturated flow scenarios and find that (*a*) the performance of IES relying on the Richards' equation is significantly
impacted by soil heterogeneity, initial, and boundary conditions, and (*b*) IES based on either ANN or Richards' equation can be notably affected by the quality of the measurements.

In this broad framework, it is noted that the accuracy of parameter estimation for a given environmental system is jointly determined by the ability of the mathematical model to describe the system of interest (Sakov et al., 2018; Alfonzo and Oliver, 2019; Luo, 2019; Evensen, 2019), the ability of the used assimilation algorithm (Emerick and Reynolds, 2013;
Bocquet and Sakov, 2014) as well as by the quantity and quality of available observations (Zha et al., 2019; Xia et al., 2018 and references therein).

With reference to a groundwater system, data which are commonly collected in a borehole and then employed for parameter estimation include head (water level or pressure), solute concentration, and/or in some cases fluxes. A well screen opened at multiple depths can provide information associated with preferential pathways of flow and/or solute transport.





Hydraulic heads observed in such a setting can be considered to constitute an integrated type of information and to be representative of an average system state (Ecli et al., 2001; 2003; Konikow et al., 2009; Zhang et al., 2018). Ecli et al. (2001; 2003) conclude that the use of long-screen wells to collect measurements should be approached with caution, as these can yield misleading and ambiguous information concerning, e.g., hydraulic head, solute concentration, location of contaminant source, and plume geometry. These types of monitoring wells can be found in a variety of field settings where head and/or

solute concentration data are collected (see, e.gs., Ecli et al., 2001; 2003; Post et al., 2007; Konikow et al., 2009; Zhang et al., 2019 and references therein). As an alternative, a somehow localized information could be provided through the use of packers. Installing the latter can be costly and in some cases impractical.

Here, we aim at exploring the effect that assimilating hydraulic head information collected over time within wells equipped with screens of differing lengths can have on our ability to characterize the spatial distribution of conductivity of a

three-dimensional fully saturated heterogeneous aquifer. We consider multi-node wells (Knoikow et al., 2009) to represent observation boreholes which can be (*a*) equipped with packers to mimic point-like measurements, (*b*) fully screened, or (*c*) partially penetrating. To this end, we focus on a convergent flow scenario driven by a partially penetrating pumping well operating in a three-dimensional heterogeneous conductivity field. Hydraulic head information is collected at a network of multi-node wells, to represent data associated with screened intervals of differing lengths along the vertical. We consider

synthetic scenarios to provide transparent comparative analyses of the extent at which the quality of the estimated conductivity fields is influenced by the type of multi-node wells considered.

Data assimilation is performed upon relying on an EnKF coupled with stochastic moment equations (MEs) of transient groundwater flow (e.g., Tartakovsky and Neuman, 1998; Zhang, 2002; Ye, et al., 2004). The latter are approximated at second order (in terms of the standard deviation of the natural logarithm of hydraulic conductivity) and are solved by an

efficient numerical scheme proposed in this study.

While we refer to Zhang (2002) and Winter et al. (2003) for reviews of MEs in heterogeneous conductivity fields, we recall that MEs of groundwater flow have been previously incorporated into geostatistical inverse modeling approaches (e.g., Hernandez et al., 2003, 2006) or stochastic pumping test interpretation (Neuman et al., 2004, 2007), and have been considered in field settings (Riva et al., 2009; Bianchi Janetti et al., 2010; Panzeri et al., 2015). More recent developments

have allowed embedding stochastic MEs of steady-state groundwater flow in model reduction strategies (Xia et al., 2020). MEs of transient groundwater flow have also been framed in the context of data assimilation/ parameter estimation approaches based on the EnKF approach (Li and Tchelepi, 2006; Panzeri et al., 2013; 2014).

Panzeri et al. (2013, 2014, 2015) present an approach for data assimilation (hereafter termed MEs-EnKF) which relies on embedding MEs of groundwater flow within an EnKF framework. They (*a*) demonstrate that the method does not suffer

from spurious correlation, thus avoiding resorting to any localization or inflation techniques, (*b*) document the computational feasibility and accuracy of the approach in two-dimensional synthetic log-conductivity domains, and then (*c*) explore a first field application to estimate log-transmissivity through assimilation of drawdown data collected during a series of cross-hole pumping tests.





An aspect which is still somehow limiting the advantages of MEs-EnKF is related to the formulation of MEs in terms of a

Green's function approach (see also Ye et al., 2004). One is then required to solve the equation satisfied by a (zero-order mean) Green's function for each node of the numerical grid employed to discretize the computational domain. While one can take advantage of symmetries related to the evaluation of the Green's function, Panzeri et al. (2014) show that in their illustrative examples the CPU time required by MEs-EnKF is equivalent to performing a classical EnKF relying on a collection of 35,000 Monte Carlo (MC) realizations. The negative impact of this computational scheme could be aggravated

in three-dimensional scenarios. Here, we circumvent this issue by solving MEs for three-dimensional transient groundwater flow by relying on the (second-order accurate) approximations of MEs presented by Zhang (2002).

The remainder of the work is structured as follows. Section 2 details the main elements associated with the mathematical background of MEs and multi-node wells. Section 3 introduces the coupling between MEs and the EnKF approach. Section 4 illustrates the synthetic settings we analyze together with the criteria according to which the performance of MEs-EnKF

and the standard Monte Carlo-based EnKF (MC-EnKF) is assessed. Section 5 is devoted to the presentation and analysis of the key results. Main conclusions of this work are presented in Section 6.

## 2 Theoretical Background

### 2.1 Stochastic moment equations for groundwater flow

We consider transient groundwater flow in a three-dimensional domain $\Omega$ described by

$$S_s(\mathbf{x})\frac{\partial h(\mathbf{x},t)}{\partial t}+\nabla_x\cdot\mathbf{q}(\mathbf{x},t)=f(\mathbf{x},t) \quad \text{with} \quad \mathbf{q}(\mathbf{x},t)=-K(\mathbf{x})\nabla h(\mathbf{x},t), \tag{1}$$

subject to initial and boundary conditions

$$h(\mathbf{x},t_0)=H_0(\mathbf{x}) \qquad\qquad\qquad \mathbf{x}\in\Omega \tag{2}$$

$$h(\mathbf{x},t)=H(\mathbf{x},t) \qquad\qquad\qquad \mathbf{x}\in\Gamma_D \tag{3}$$

$$\left[-\mathbf{q}(\mathbf{x},t)\right]\cdot\mathbf{n}(\mathbf{x})=Q(\mathbf{x},t) \qquad\qquad \mathbf{x}\in\Gamma_N \tag{4}$$

where $\mathbf{x}$ denotes the vector of Euclidian coordinates; $t$ is time; $K$ is hydraulic conductivity; $S_s$ is specific storage; $h$ is hydraulic head; $\mathbf{q}$ is Darcy flux; $f$ is a forcing term; $H_0(\mathbf{x})$ is initial hydraulic head; $H(\mathbf{x},t)$ is head along the Dirichlet boundary; and $Q$ is a prescribed flux along the Neuman boundary. In the following, we consider $S_s(\mathbf{x})$, $H(\mathbf{x},t)$ and $Q(\mathbf{x},t)$ as deterministic, while $H_0(\mathbf{x})$, $f(\mathbf{x},t)$ and $K(\mathbf{x})$ are taken to be random quantities.

The natural logarithm of hydraulic conductivity, $Y(\mathbf{x})=\ln K(\mathbf{x})$, is assumed to be a second-order stationary process

correlated in space with mean $\langle Y(\mathbf{x})\rangle$ and variance $\sigma_Y^2$. Tartakovsky and Neuman (1998a, b) derive integro-differential MEs to compute space-time dynamics of (ensemble) means and covariances of hydraulic heads and fluxes. They then resort



to a perturbation approach to derive recursive approximations of these otherwise exact integro-differential MEs. Ye et al. (2004) solve second-order (in the standard deviation of $Y$, $\sigma_Y$) approximations of these MEs by finite elements for superimposed mean-uniform and convergent flows for two-dimensional settings. Since numerical solutions of moment

equations are heavy in terms of computational resources (Zhang, 2002; Ye et al., 2004), in the following subsections we illustrate a workflow which enables us to evaluate all quantities of interest (up to second order in $\sigma_Y$) with reduced computational efforts.

### 2.1.1 Mean head and flux

We start by expressing a given random quantity, $\Im$, as the sum of its (ensemble) mean, $\langle \Im \rangle$, and a zero-mean random

fluctuation, $\Im'$. Here and in the following mean head and flux are approximated up to second order in $\sigma_Y$ as

$$\langle h(\mathbf{x},t)\rangle \approx \langle h^{(0)}(\mathbf{x},t)\rangle + \langle h^{(2)}(\mathbf{x},t)\rangle ; \quad \langle \mathbf{q}(\mathbf{x},t)\rangle \approx \langle \mathbf{q}^{(0)}(\mathbf{x},t)\rangle + \langle \mathbf{q}^{(2)}(\mathbf{x},t)\rangle , \tag{5}$$

and the following equations hold

$$\begin{cases} S_s(\mathbf{x})\dfrac{\partial \langle h^{(0)}(\mathbf{x},t)\rangle}{\partial t} + \nabla_x \cdot \langle \mathbf{q}^{(0)}(\mathbf{x},t)\rangle = \langle f^{(0)}(\mathbf{x},t)\rangle & \mathbf{x}\in\Omega \\[2mm] \langle \mathbf{q}^{(0)}(\mathbf{x},t)\rangle = -K_G(\mathbf{x})\nabla\langle h^{(0)}(\mathbf{x},t)\rangle & \mathbf{x}\in\Omega \\[2mm] \langle h^{(0)}(\mathbf{x},t)\rangle = \langle H_0^{(0)}(\mathbf{x})\rangle & \mathbf{x}\in\Omega \\[2mm] \langle h^{(0)}(\mathbf{x},t)\rangle = H(\mathbf{x},t) & \mathbf{x}\in\Gamma_D \\[2mm] -\langle \mathbf{q}^{(0)}(\mathbf{x},t)\rangle \cdot \mathbf{n}(\mathbf{x}) = Q(\mathbf{x},t) & \mathbf{x}\in\Gamma_N \end{cases} , \tag{6}$$

$$\begin{cases} S_s(\mathbf{x})\dfrac{\partial \langle h^{(2)}(\mathbf{x},t)\rangle}{\partial t} + \nabla_x \cdot \langle \mathbf{q}^{(2)}(\mathbf{x},t)\rangle = \langle f^{(2)}(\mathbf{x},t)\rangle & \mathbf{x}\in\Omega \\[2mm] \langle \mathbf{q}^{(2)}(\mathbf{x},t)\rangle = -K_G(\mathbf{x})\left(\nabla_x\langle h^{(2)}(\mathbf{x},t)\rangle + \dfrac{\sigma_Y^2}{2}\nabla_x\langle h^{(0)}(\mathbf{x},t)\rangle\right) + \mathbf{r}(\mathbf{x},t) & \mathbf{x}\in\Omega \\[2mm] \langle h^{(2)}(\mathbf{x},t)\rangle = \langle H_0^{(2)}(\mathbf{x})\rangle & \mathbf{x}\in\Omega \\[2mm] \langle h^{(2)}(\mathbf{x},t)\rangle = 0 & \mathbf{x}\in\Gamma_D \\[2mm] -\langle \mathbf{q}^{(2)}(\mathbf{x},t)\rangle \cdot \mathbf{n}(\mathbf{x}) = 0 & \mathbf{x}\in\Gamma_N \end{cases} \tag{7}$$

Here, superscript ($i$) indicates terms that are strictly of order $i$ (in terms of powers of $\sigma_Y$), $K_G(\mathbf{x}) = e^{\langle Y(\mathbf{x})\rangle}$ is the geometric mean of $K(\mathbf{x})$, and $\mathbf{r}(\mathbf{x},t)$ is the second-order residual flux evaluated as $\mathbf{r}(\mathbf{x},t) = \lim\limits_{y\to x}\left[-\nabla_x u(\mathbf{y},\mathbf{x},t)\right]$ (e.g., Xia et al., 2019





and references therein), where $u(\mathbf{y},\mathbf{x},t) = \langle K'(\mathbf{y})h'(\mathbf{x},t)\rangle^{(2)}$ is the second-order approximation of the cross-covariance between hydraulic head and conductivity, computed as detailed in Section 2.1.2.

### 2.1.2 Cross-Covariance between hydraulic head and conductivity

Multiplying Eqs. (1)-(4) by $K'(\mathbf{y})$ and taking expectation yield the following equation governing the evolution of $u(\mathbf{y},\mathbf{x},t)$

$$
\begin{cases}
S_s(\mathbf{x})\dfrac{\partial u(\mathbf{y},\mathbf{x},t)}{\partial t} = \\
\quad = \nabla_x \cdot \left[ K_G(\mathbf{x})\nabla_x u(\mathbf{y},\mathbf{x},t) - K_G(\mathbf{y})C_Y(\mathbf{x},\mathbf{y})\langle \mathbf{q}^{(0)}(\mathbf{x},t)\rangle \right] + C_{fK}(\mathbf{y},\mathbf{x},t) & \mathbf{x}\in\Omega \\
u(\mathbf{y},\mathbf{x},t_0) = U_0(\mathbf{y},\mathbf{x}) & \mathbf{x}\in\Omega \\
u(\mathbf{y},\mathbf{x},t) = 0 & \mathbf{x}\in\Gamma_D \\
\left[ K_G(\mathbf{x})\nabla_x u(\mathbf{y},\mathbf{x},t) - K_G(\mathbf{y})C_Y(\mathbf{x},\mathbf{y})\langle \mathbf{q}^{(0)}(\mathbf{x},t)\rangle \right]\cdot\mathbf{n}(\mathbf{x}) = 0 & \mathbf{x}\in\Gamma_N
\end{cases}
\tag{8}
$$

Here, $C_Y(\mathbf{x},\mathbf{y}) = \langle Y'(\mathbf{x})Y'(\mathbf{y})\rangle$ is the covariance of the log-conductivity field, $C_{fK}(\mathbf{y},\mathbf{x},t) = \langle K'(\mathbf{y})f'(\mathbf{x},t)\rangle^{(2)}$ is the second-order cross-covariance between conductivity $K(\mathbf{y})$ and forcing term $f(\mathbf{x},t)$, and $U_0(\mathbf{y},\mathbf{x}) = \langle K'(\mathbf{y})H_0'(\mathbf{x})\rangle^{(2)}$ is the second-order approximation of the cross-covariance between $K$ and initial hydraulic head. Note that $U_0(\mathbf{y},\mathbf{x})$ vanishes 150   when $H_0$ is deterministic.

### 2.1.3 Head covariance

The equation governing the evolution of the (second-order) head covariance between space-time locations $(\mathbf{y},\tau)$ and $(\mathbf{x},t)$, $C_h(\mathbf{y},\mathbf{x},\tau,t) = \langle h'(\mathbf{y},\tau)h'(\mathbf{x},t)\rangle^{(2)}$, is given by (Zhang, 2002)

$$
\begin{cases}
S_s(\mathbf{x})\dfrac{\partial C_h(\mathbf{y},\mathbf{x},\tau,t)}{\partial t} = \\
\quad = \nabla_x \cdot \left[ K_G(\mathbf{x})\nabla_x C_h(\mathbf{y},\mathbf{x},\tau,t) + u(\mathbf{x},\mathbf{y},\tau)\nabla_x\langle h^{(0)}(\mathbf{x},t)\rangle \right] + C_{fh}(\mathbf{y},\mathbf{x},\tau,t) & \mathbf{x}\in\Omega \\
C_h(\mathbf{y},\mathbf{x},\tau,t_0) = \langle h'(\mathbf{y},\tau)h'(\mathbf{x},t_0)\rangle^{(2)} & \mathbf{x}\in\Omega \\
C_h(\mathbf{y},\mathbf{x},\tau,t) = 0 & \mathbf{x}\in\Gamma_D \\
\left[ K_G(\mathbf{x})\nabla_x C_h(\mathbf{y},\mathbf{x},\tau,t) + u(\mathbf{x},\mathbf{y},\tau)\nabla_x\langle h^{(0)}(\mathbf{x},t)\rangle \right]\cdot\mathbf{n}(\mathbf{x}) = 0 & \mathbf{x}\in\Gamma_N
\end{cases}
\tag{9}
$$

where $u(\mathbf{x},\mathbf{y},\tau) = \langle K'(\mathbf{x})h'(\mathbf{y},\tau)\rangle^{(2)}$ is given by Eq. (8), and $C_{fh}(\mathbf{y},\mathbf{x},\tau,t) = \langle h'(\mathbf{y},\tau)f'(\mathbf{x},t)\rangle^{(2)}$ is the second-order cross-covariance between forcing term $f(\mathbf{x},t)$ and hydraulic head $h(\mathbf{y},\tau)$. To minimize redundancy, hereinafter we omit stating



that all cross-/auto- covariances of quantities of interest appearing in our formulations are to be considered as second-order approximations.

## 2.2 Monitoring wells

We consider three kinds of observation wells, leading to three diverse types of hydraulic head information (see Fig. 1). Type A wells are characterized by packers located at three depths, where point-wise hydraulic head observations are collected. Otherwise, Type B and/or C wells represent partially and fully penetrating wells, respectively, and provide hydraulic head values that are averaged along the corresponding screened intervals. Note that, even though there is no pumping from B- and C-wells, there are flux exchanges between these wells and the surrounding aquifer system, as opposed to the setting

associated with packers (A-wells). Such flow is related to the difference between the water level within the well and hydraulic head values along the borehole.

Following Konokow et al. (2009), neglecting linear (due to skin effects) and non-linear (due to turbulent flow) well loss terms, the water level at well $I$, $h_I^w$, (Type B and/or C) at a given time $t$ (omitted in the following equations for brevity) can be evaluated through

$$h_I^w = \frac{\sum_{i=1}^n b_i K_i h_i}{\sum_{i=1}^n b_i K_i} \tag{10}$$

where $n$ is the number of nodes in the multi-node observation well $I$, i.e., the number of cells according to which the well screen is discretized; $h_i$, $b_i$, and $K_i$ are the hydraulic head, thickness and conductivity of the cell of the numerical grid whose centroid corresponds to the $i^{th}$ node in the multi-node well, respectively. Note that Eq. (10) has been derived assuming that the flux exchange, i.e., the flow into (or out of) the monitoring well at the $i^{th}$ node, $Q_i$, depends linearly on the product $b_i K_i$

(see also Section 2.2.2). Numerical evaluation of $h_i$ at a given time $t$ requires evaluating $Q_i$, as shown in Section 2.2.2.

### 2.2.1 Moments for hydraulic head at observation wells

Mean head at well $I$ is approximated (at second order in $\sigma_Y$) as $\langle h_I^w \rangle \approx \langle h_I^{w(0)} \rangle + \langle h_I^{w(2)} \rangle$ where, starting from Eq. (10), one can obtain the zero-, $\langle h_I^{w(0)} \rangle$, and second-, $\langle h_I^{w(2)} \rangle$, order components as

$$\left\langle h_I^{w(0)} \right\rangle = \frac{\sum_{i=1}^n T_{G,i} \left\langle h_i^{(0)} \right\rangle}{\sum_{i=1}^n T_{G,i}} \tag{11}$$

$$\left\langle h_I^{w(2)} \right\rangle \sum_{i=1}^n T_{G,i} = \sum_{i=1}^n T_{G,i} \left\{ \frac{u_{ii}}{K_{G,i}} - \frac{\left\langle K_i' h_I'^{rw} \right\rangle^{(2)}}{K_{G,i}} + \left\langle h_i^{(2)} \right\rangle + \frac{\sigma_{Y,i}^2}{2} \left( \left\langle h_i^{(0)} \right\rangle - \left\langle h_I^{w(0)} \right\rangle \right) \right\} \tag{12}$$



Here, $T_{G,i} = b_i K_{G,i}$, $\langle h_i^{(0)} \rangle$, $\langle h_i^{(2)} \rangle$, $K_{G,i}$ and $\sigma_{Y,i}^2$ correspond to the zero- (evaluated by Eq. (6)) and the second- (evaluated by Eq. (7)) order mean head, geometric mean of conductivity and variance of log-conductivity at the $i^{th}$ cell of a multi-node monitoring well, respectively; $u_{ii} = \langle K_i' h_i' \rangle^{(2)}$ is the cross-covariance between conductivity and head at the $i^{th}$ cell; $\langle K_i' h_I'^w \rangle^{(2)}$ is the cross-covariance between well head and conductivity at the $i^{th}$ cell (evaluated as detailed below, see Eq.

185 (15)).

The covariance between water levels at wells $I$ (i.e., $h_I^w$) and $J$ (i.e., $h_J^w$), $C_{h^w} = \langle h_I'^w h_J'^w \rangle^{(2)}$, can be evaluated as (see also Appendix A, Eqs. (A1)-(A3))

$$
C_{h^w} \sum_{i=1}^{n} T_{G,i} \sum_{j=i}^{m} T_{G,j} =
$$

$$
= \sum_{j=1}^{m} \sum_{i=1}^{n} T_{G,i} T_{G,j}
\left\{
\begin{aligned}
&\left( \left( \langle h_i^{(0)} \rangle - \langle h_I^{w(0)} \rangle \right) \left( \langle h_j^{(2)} \rangle - \langle h_J^{w(2)} \rangle + \frac{u_{jj}}{K_{G,j}} + \frac{u_{ij}}{K_{G,i}} \right) \right.\\
&+ \left( \langle h_j^{(0)} \rangle - \langle h_J^{w(0)} \rangle \right) \left( \langle h_i^{(2)} \rangle - \langle h_I^{w(2)} \rangle + \frac{u_{ji}}{K_{G,j}} + \frac{u_{ii}}{K_{G,i}} \right) \\
&+ \left( \frac{\sigma_{Y,i}^2}{2} + \frac{\sigma_{Y,j}^2}{2} + C_{Y,ij} \right) \left( \langle h_i^{(0)} \rangle - \langle h_I^{w(0)} \rangle \right) \left( \langle h_j^{(0)} \rangle - \langle h_J^{w(0)} \rangle \right) \\
&+ C_{h,ij}
\end{aligned}
\right\}
\tag{13}
$$

where $T_{G,i} = b_i K_{G,i}$, $T_{G,j} = b_j K_{G,j}$, $I$ and $J$ are indices ranging from 1 to $N_w$ (i.e., the total number of monitoring wells of

Type B and C); $n$ and $m$ correspond to the total number of cells according to which the screens of boreholes $I$ and $J$ are discretized, respectively; $C_{Y,ij}$ is the log-conductivity covariance between the $i^{th}$ and $j^{th}$ cells of boreholes $I$ and $J$, respectively; $u_{ji}$ (or $u_{ij}$) is the cross-covariance between the conductivity of the $j^{th}$ cell of well $J$ (or the $i^{th}$ cell of well $I$) and head of the $i^{th}$ cell of well $I$ (or the $j^{th}$ cell of well $J$); $C_{h,ij}$ is the head covariance between the $i^{th}$ cell of well $I$ and the $j^{th}$ cell of well $J$. Terms $u_{ji}$ (or $u_{ij}$) and $C_{h,ij}$ can be readily obtained by solving Eqs. (8)-(9).

The cross-covariance between $h_I^w$ at a given time $t$ and aquifer head $h$ at time $\tau$ (at a given location, omitted for brevity), i.e., $C_{h_I^w h^\tau} = \langle h_I'^w(t) h'(\tau) \rangle^{(2)}$ is given by

$$
C_{h_I^w h^\tau} \sum_{i=1}^{n} T_{G,i} = \sum_{i=1}^{n} T_{G,i} \left[ C_{h,i,\tau} + \frac{u_{i,\tau}}{K_{G,i}} \left( \langle h_i^{(0)} \rangle - \langle h_I^{w(0)} \rangle \right) \right]
\tag{14}
$$


Here, $u_{i,\tau} = \left\langle K_i' \, h'(\tau) \right\rangle^{(2)}$ represents the cross-covariance between conductivity at the $i^{th}$ cell of well $I$ and aquifer head (at a

given location) at time $\tau$. Likewise, $C_{h,i,\tau} = \left\langle h_i'(t) h'(\tau) \right\rangle^{(2)}$ represents head covariance between head at time $t$ at the $i^{th}$ cell of

the monitoring well $I$ and head at time $\tau$ at a given location in the aquifer.

The cross-covariance between $h_I^w$ at time $t$ and conductivity at a given location in the aquifer, $C_{h_I^w K} = \left\langle h_I'^w(t) K' \right\rangle^{(2)}$, can be

expressed as

$$C_{h_I^w K} \sum_{i=1}^n T_{G,i} = \sum_{i=1}^n T_{G,i} \left[ u_i + K_G C_{Y,i} \left( \left\langle h_i^{(0)} \right\rangle - \left\langle h_I^{w(0)} \right\rangle \right) \right] \tag{15}$$

where $C_{Y,i} = \left\langle Y_i' Y' \right\rangle$ is the covariance between log-conductivity at the $i^{th}$ cell along the monitoring borehole $I$ and log-

conductivity at a given location in the domain, and $u_i = \left\langle K' \, h_i'(t) \right\rangle^{(2)}$ is cross-covariance between conductivity at a given

point in the domain and aquifer head at time $t$ at the $i^{th}$ cell along the monitoring borehole $I$.

It is worthwhile to note that covariances and cross-covariances evaluated in Eqs. (13)-(15) depend explicitly on the
difference between the mean water level at the monitoring well and the mean hydraulic head along the well screen.

### 2.2.2 Moments for flux between a monitoring well and the aquifer

Assuming that the evolution of head at the observation borehole $I$ can be conceptualized as a sequence of temporal events
(each associated with the attainment of instantaneous equilibrium conditions). The link between $h_I^w$, $h_i$, and $Q_i$ can then be
obtained by relying on the Thiem (1906) equation as

$$h_I^w = h_i + \frac{Q_i}{a \, b_i K_i}, \qquad \text{with } a = \frac{2\pi}{\ln\left(r_0 / r_w\right)} \tag{16}$$

where $r_0$ and $r_w$ are the effective (e.g., the radius of the well that would give the same head as that calculated at the node of

the cell that contains the well) and actual well radius, respectively. The mean flux exchange is approximated as

$\left\langle Q_i \right\rangle \approx \left\langle Q_i^{(0)} \right\rangle + \left\langle Q_i^{(2)} \right\rangle$ and from Eq. (16) one can write

$$\left\langle Q_i^{(0)} \right\rangle = aT_{G,i} \left( \left\langle h_I^{w(0)} \right\rangle - \left\langle h_i^{(0)} \right\rangle \right); \tag{17}$$

$$\left\langle Q_i^{(2)} \right\rangle = aT_{G,i} \left\{ \left\langle h_I^{w(2)} \right\rangle - \left\langle h_i^{(2)} \right\rangle + \frac{\sigma_{Y,i}^2}{2} \left( \left\langle h_I^{w(0)} \right\rangle - \left\langle h_i^{(0)} \right\rangle \right) + \frac{\left\langle h_I'^w K_i' \right\rangle^{(2)}}{K_{G,i}} - \frac{\left\langle h_i' K_i' \right\rangle^{(2)}}{K_{G,i}} \right\} \tag{18}$$

The cross-covariance between $Q_i$ at time $t$ and aquifer head at time $\tau$, $C_{Q_i h^\tau} = \left\langle Q_i'(t) h'(\tau) \right\rangle^{(2)}$, can be expressed as

$$C_{Q_i h^\tau} = aT_{G,i} \left\{ C_{h_I^w h^\tau} - C_{h_i h^\tau} + \frac{u_{i,\tau}}{K_{G,i}} \left( \left\langle h_I^{w(0)} \right\rangle - \left\langle h_i^{(0)} \right\rangle \right) \right\} \tag{19}$$





where $C_{h_I^{'w} h^{\tau}} = \left\langle h_I^{'w}(t) h'(\tau) \right\rangle^{(2)}$, $C_{h_i h^{\tau}} = \left\langle h_i'(t) h'(\tau) \right\rangle^{(2)}$. Finally, one can obtain the following expression for the cross-covariance between $Q_i$ and $K$, $C_{Q_i K} = \left\langle Q_i' K' \right\rangle^{(2)}$,

$$C_{Q_i K} = a T_{G,i} \left\{ C_{h_I^w K} - u_i + K_G C_{Y,i} \left( \left\langle h_I^{w(0)} \right\rangle - \left\langle h_i^{(0)} \right\rangle \right) \right\} \tag{20}$$

**2.3 Numerical solution strategy**

We solve numerically Eqs. (6)-(9) by approximating the spatial derivatives through a finite element approach and the temporal derivatives through an implicit method. As in Xia et al. (2019), moments $\left\langle h^{(0)} \right\rangle$, $u$, and $\left\langle h^{(2)} \right\rangle$ are sequentially obtained by solving Eqs. (6), (8), and (7), respectively. Details associated with the evaluation of $C_h$, which requires $\left\langle h^{(0)} \right\rangle$ and $u$ as inputs, are illustrated in the following.

For the purpose of our data assimilation workflow, we start by noting that we are interested in computing $C_h$ associated

with two identical time coordinates, i.e., $C_h(\mathbf{y}, \mathbf{x}, \tau=t, t) = \left\langle h'(\mathbf{y}, \tau=t) h'(\mathbf{x}, t) \right\rangle^{(2)}$. We then recall that Zhang (2002) computes $C_h(\mathbf{y}, \mathbf{x}, \tau=t, t)$ for each time $t$ (while $C_h(\mathbf{y}, \mathbf{x}, \tau=t, t-\Delta t)$ is also unknown, $\Delta t$ being a constant temporal step size) by solving for $C_h(\mathbf{y}, \mathbf{x}, \tau=t, t')$ from $t' = 0$ to $t' = t$. While this procedure can be computationally heavy for long times, Zhang (2002) points out that when flow changes only mildly, $C_h(\mathbf{x}, \mathbf{y}, \tau = t, t-\Delta t) \approx C_h(\mathbf{x}, \mathbf{y}, \tau = t-\Delta t, t-\Delta t)$, an approximation whose general validity is still not completely explored.

Here, we circumvent this issue and obtain high computational efficiency by directly evaluating $C_h(\mathbf{y}, \mathbf{x}, \tau=t, t)$ from $C_h(\mathbf{y}, \mathbf{x}, \tau=t-\Delta t, t-\Delta t)$ via (i) computing $C_h(\mathbf{y}, \mathbf{x}, \tau=t, t-\Delta t)$ through the solution of the equation obtained by considering Eq. (9) where the space and time derivatives operate on $\tau$ and $\mathbf{y}$ (instead of $t$ and $\mathbf{x}$) from time $t - \Delta t$ to $t$ using $C_h(\mathbf{y}, \mathbf{x}, \tau=t-\Delta t, t-\Delta t)$ as initial condition and then (ii) assessing $C_h(\mathbf{y}, \mathbf{x}, \tau=t, t)$ by solving Eq. (9) using $C_h(\mathbf{y}, \mathbf{x}, \tau=t, t-\Delta t)$ as initial condition.

It is further noted that Eqs. (6)-(9) are characterized by the same format, their discretization leading to a system of equations where the coefficients of the unknown quantities are identical, the corresponding right-hand-side terms (i.e., the forcing terms) being a function of the (ensemble) moment to be solved. In this context, one can resort to a direct solver for each time step. Thus, factorization of the matrix containing the coefficients of the system of equations is performed only once, resulting in a high computational efficiency because only the right-hand-side term needs to be updated, depending on

the moment of interest.





With reference to the forcing terms $\langle f^{(0)} \rangle$, $\langle f^{(2)} \rangle$, $C_{fK}$, and $C_{fh}$ in Eqs. (6)-(9), we note that these vanish for Type A wells and when one disregards flux exchanges between Type B (or C) wells and the aquifer. In these instances, mean head values and the associated covariance are simply obtained upon evaluating numerically Eqs. (6)-(9). Thus, when considering a time interval $[t - \Delta t, t]$, the main computational cost stems from the evaluation of $u(\mathbf{y}, \mathbf{x}, t)$, $C_h(\mathbf{y}, \mathbf{x}, \tau=t, t-\Delta t)$, and

$C_h(\mathbf{y}, \mathbf{x}, \tau=t, t)$, each of these requiring $N$ times the computational cost (hereafter denoted as $C_c^{\mathrm{MEs}}$) associated with the solution of the system of $N$ equations resulting after discretization. Therefore, the total computational effort required for solving Eqs. (6)-(9) at each time step is $3N C_c^{\mathrm{MEs}}$. Note that the computational effort is reduced to $2N C_c^{\mathrm{MEs}}$ for the first time interval, when the initial head is deterministic, or for a steady-state flow scenario (see Xia et al., 2019).

Otherwise, considering flux exchange processes when representing Type B (or C) wells entails evaluation of the source

terms in Eqs. (6)-(9) as $\langle f^{(0)} \rangle = \langle Q_i^{(0)} \rangle$, $\langle f^{(2)} \rangle = \langle Q_i^{(2)} \rangle$, $C_{fk} = C_{Q_i K}$, and $C_{fh} = C_{Q_i h^\tau}$. The evaluation of the (ensemble) moments of interest across time interval $[t - \Delta t, t]$ is then performed through the workflow depicted in Fig. 2. In this case, we note that convergence of the iterative procedure is attained when the absolute difference between mean well heads at iteration $iter+1$, $\langle h_I^w \rangle_{iter+1}$, and $iter$, $\langle h_I^w \rangle_{iter}$, is lower than a pre-set value $\varepsilon$. The main computational effort required for these evaluations corresponds to $3(iter+1)N C_c^{\mathrm{MEs}}$ for each time step.

## 3 Ensemble Kalman Filter coupled with moment equations

We start by introducing the mean system state vector $\langle \boldsymbol{\varphi} \rangle$ as

$$\langle \boldsymbol{\varphi} \rangle = \begin{bmatrix} \langle \mathbf{h} \rangle & \langle \mathbf{h}^w \rangle & \langle \mathbf{Y} \rangle \end{bmatrix}^T \tag{21}$$

where $\langle \mathbf{h} \rangle$, $\langle \mathbf{Y} \rangle$ correspond, respectively, to $N$-dimensional vectors of mean head and mean log-conductivity, and $\langle \mathbf{h}^w \rangle$ is a $N_w$-dimensional mean well head vector, subscript $T$ representing transpose.

Each data assimilation cycle, corresponding to time interval $[t - \Delta t, t]$ comprises a forecast (or forward propagation) step and an update (or analysis) step. The forecast step is implemented by solving the moment equations described in Section 2. We write the predicted mean and covariance of the system state as

$$\langle \boldsymbol{\varphi} \rangle^f = \begin{bmatrix} \langle \mathbf{h} \rangle^f & \langle \mathbf{h}^w \rangle^f & \langle \mathbf{Y} \rangle \end{bmatrix}^T ; \quad \mathbf{P}^f = \begin{bmatrix} \mathbf{C}_h^f & \left[\mathbf{C}_{h^w h}^f\right]^T & \left[\mathbf{C}_{Yh}^f\right]^T \\ \mathbf{C}_{h^w h}^f & \mathbf{C}_{h^w}^f & \left[\mathbf{C}_{Yh^w}^f\right]^T \\ \mathbf{C}_{Yh}^f & \mathbf{C}_{Yh^w}^f & \mathbf{C}_Y^f \end{bmatrix} \tag{22}$$





Here, superscript $f$ represents predicted quantities obtained in the forecast step, $\langle \mathbf{h} \rangle^f$ is the predicted mean head (Eq. 5),

$\langle \mathbf{h}^w \rangle^f$ is the predicted mean water level at monitoring borehole (Eqs. 11-12), $\langle \mathbf{Y} \rangle$ is the updated natural logarithm of

conductivity obtained at the previous data assimilation cycle, $\mathbf{C}_h^f$ is the predicted $N \times N$-dimensional head covariance

matrix, (Eq. 9), $\mathbf{C}_{h^w h}^f$ is the $N_w \times N$-dimensional predicted cross-covariance between well and aquifer head (Eq. 14), $\mathbf{C}_{h^w}^f$ is

the predicted $N_w \times N_w$-dimensional covariance of well head (Eq. 13), $\mathbf{C}_{Yh}^f$ is the predicted $N \times N$-dimensional cross-

covariance between $Y$ and aquifer head $h$ (Eq. 8), $\mathbf{C}_{Yh^w}^f$ is the predicted $N \times N_w$-dimensional cross-covariance between $Y$ and

well head $h^w$ (Eq. 15), $\mathbf{C}_Y^f$ is $N \times N$-dimensional $Y$ covariance and is equal to its updated counterpart associated with the

previous updating step.

The equations used to evaluate the state updated vector $\langle \boldsymbol{\varphi} \rangle^{up}$ and the updated covariance matrix $\mathbf{P}^{up}$ are

$$\langle \boldsymbol{\varphi} \rangle^{up} = \langle \boldsymbol{\varphi} \rangle^f + \mathbf{C}_{\varphi d} \left( \mathbf{C}_{dd} + \alpha \mathbf{C}_D \right)^{-1} \left( \langle \boldsymbol{\varphi}_{so} \rangle - \mathbf{d}_{obs} \right) \tag{23}$$

and

$$\mathbf{P}^{up} = \left( \mathbf{I} - \mathbf{C}_{\varphi d} \left( \mathbf{C}_{dd} + \alpha \mathbf{C}_D \right)^{-1} \mathbf{H} \right) \mathbf{P}^f \tag{24}$$

where $\mathbf{C}_{\varphi d}$ $(= \mathbf{P}^f \mathbf{H}^T)$ is the $(2N + N_w) \times d$-dimensional cross-covariance between the system state and the simulated

observations, matrix $\mathbf{H}$ of dimension $d \times (2N + N_w)$ is the observation operator that describes the relationship between the

system state and the observations, $\mathbf{C}_{dd}$ $(= \mathbf{H} \mathbf{P}^f \mathbf{H}^T)$ is the $d \times d$-dimensional covariance matrix of the simulated observations,

$\mathbf{C}_D$ is the $d \times d$-dimensional covariance of observation errors, $\mathbf{I}$ is the identity matrix, $\alpha$ is a constant inflation factor, $\alpha = 1$

corresponding to the uninflated Ensemble Kalman filter, $\langle \boldsymbol{\varphi}_{so} \rangle$ is the mean vector of the simulated observations, and $\mathbf{d}_{obs}$ is

the $d$-dimensional observation vector.

After the update step, $\langle \boldsymbol{\varphi} \rangle^{up}$ and $\mathbf{P}^{up}$ are expressed as

$$\langle \boldsymbol{\varphi} \rangle^{up} = \begin{bmatrix} \langle \mathbf{h} \rangle^{up} & \langle \mathbf{h}^w \rangle^{up} & \langle \mathbf{Y} \rangle^{up} \end{bmatrix}^T; \qquad \mathbf{P}^{up} = \begin{bmatrix} \mathbf{C}_h^{up} & \left[ \mathbf{C}_{h^w h}^{up} \right]^T & \left[ \mathbf{C}_{Yh}^{up} \right]^T \\ \mathbf{C}_{h^w h}^{up} & \mathbf{C}_{h^w}^{up} & \left[ \mathbf{C}_{Yh^w}^{up} \right]^T \\ \mathbf{C}_{Yh}^{up} & \mathbf{C}_{Yh^w}^{up} & \mathbf{C}_Y^{up} \end{bmatrix} \tag{25}$$

where all symbols have the same meaning (yet updated) as in Eq. (22).

When moving to a subsequent time interval during the assimilation process, we follow Panzeri et al. (2013) and (*i*) use the

updated mean head vector $\langle \mathbf{h} \rangle^{up}$ as the initial condition of the governing equation for the zero-order mean head, i.e., Eq. (6);



(*ii*) making use of $\langle \mathbf{Y} \rangle^{up}$, evaluate the updated geometric mean *N*-vector $\boldsymbol{K}_G^{up}$; (*iii*) obtain the initial condition of Eq. (8) through the product $\boldsymbol{K}_G^{up} \mathbf{C}_{Yh}^{up}$; (*iv*) use $\mathbf{C}_h^{up}$ as the initial condition of Eq. (9); and (*v*) use $\boldsymbol{K}_G^{up}$ and $\mathbf{C}_Y^{up}$ as inputs to Eqs. (6)-(9) and Eqs. (11)-(20).

It should be noted that, if one neglects flux exchanges between the aquifer and Type B and/or C monitoring wells (or a Type A well is considered), moments including water level at well (i.e., $\langle \mathbf{h}^w \rangle$, $\mathbf{C}_{h^w h}$, $\mathbf{C}_{h^w}$, $\mathbf{C}_{Y h^w}$) should be omitted in Eqs. (21)-(25).

## 4 Illustrative examples

We consider a three-dimensional domain (Fig. 3a) of size $600 \times 600 \times 60$ (hereafter, all quantities are considered in

consistent units), the system being discretized onto a numerical mesh comprising $25 \times 25 \times 13$ nodes, for a total of 34,560 tetrahedrons. A partially penetrating pumping well pumps at a constant rate of 1,000 for $0 \leq t \leq 30$, after which water withdrawal stops and a recovering process takes place for $30 < t \leq 60$. We subdivide the overall simulation time according to 20 uniform intervals, which can potentially be used for assimilation of head observations. The well pumping rate is uniformly distributed across the central nodes of layers no. 1 and 2 in the numerical mesh (numbering is from top to bottom

of the domain). The left and right sides of the system are set as Dirichlet boundaries, where a deterministic head $H = 100$ is fixed, the remaining boundaries being considered impervious. Initial head and storativity are deterministic and set equal to 100 and $10^{-3}$, respectively. The natural logarithm of conductivity, *Y*, is modeled as a spatially correlated second-order stationary random field with covariance given by

$$C_Y = \sigma_Y^2 \exp\left(-\left[\frac{\delta_1}{\lambda_1} + \frac{\delta_2}{\lambda_2} + \frac{\delta_3}{\lambda_3}\right]\right) \tag{26}$$

Here, $\delta_i$ and $\lambda_i$ denote, respectively, the lag and correlation scale between two points along direction $x_i$ (with $i = 1, 2, 3$).

Twenty virtual monitoring wells are regularly distributed across the domain (Fig. 3b). Type A boreholes are mimicked by considering three packers positioned at three distinct depths, corresponding to layers no. 4, 7, and 10. Type B wells are equipped with three screens (i.e., $n = 3$) whose barycenter is set at the same depths of the packers in Type A wells (see Fig. 1) and $b_i = 5$ with $i = 1, 2, 3$. Type C wells are completely penetrating across the 13 layers of the domain (i.e., $n = 13$) with

$b_1 = b_{13} = 2.5$ and $b_i = 5$ for $i = 2, \ldots, 12$.

Reference hydraulic head values which are collected at Type B and C wells and employed in the data assimilation procedure are evaluated upon solving the flow problem on the reference hydraulic conductivity fields described in the following. Flux exchanges between the aquifer and monitoring wells are evaluated according to the procedure described in Sections 2.2 and 2.3 upon setting the convergence criterion $\varepsilon = 10^{-6}$.


The effective radius of the monitoring wells is evaluated as (Chen and Zhang, 2009) $r_0 = 2^{0.25} e^{-0.75\pi} \Delta x \approx 0.113 \Delta x = 2.81$ ($\Delta x$ = 25 being the horizontal size of a given element in the computational mesh). For the purpose of our illustration example, we set $r_w = 0.1$ (i.e., $a = 1.88$).

We organize our exemplary settings according to the following four groups (for a total of 26 Test Cases, TCs) collected in Table 1.

(1) *Group* 1. It includes 7 TCs (TC1-TC7) that allow exploring the way conductivity estimates can be influenced by relying on the assimilation of head data collected at diverse types of virtual observation boreholes, while considering a simplified modeling approach where flux exchanges between the aquifer and Type B (or C) monitoring wells are neglected during the data assimilation procedure, head observations (considered in the data assimilation procedure) corresponding to depth-averaged values along the corresponding screens. We note that relying on this approach is tantamount to considering

an imperfect flow model and would possibly oversimplify the mathematical representation of the system behavior, when compared to the one employed for constructing the reference head field. Nevertheless, it has the advantage of requiring a straightforward numerical implementation.

A zero-mean reference $Y$ field is generated at the nodes of the computational mesh upon relying on the widely tested and used SGSIM code (e.g., Deutsch and Journel, 1998) by setting a unit variance, $\lambda_1 = \lambda_2 = 100$ and $\lambda_3 = 20$. While the

correlation scale values are considered as perfectly known, we aim at estimating mean and variance of $Y$. The initial guesses employed for the variance and mean of $Y$ during data assimilation are 1.0 and 0.2, respectively. Test Cases 1-3 are designed in a way that all 20 observation boreholes are of Type A, B, or C, respectively. Test Cases 4-7 enable one to explore the way conductivity estimates can depend on the use of Type A boreholes (i.e., equipped with packers) within different zones, while considering Type B or C wells in the remaining regions. Test Case 4 comprises Type A wells within zone 1 in Fig. 3b (i.e.,

within distances shorter than $\lambda_1$ from the pumping well), Type B boreholes being installed in zones 2 and 3 (at distances larger than $\lambda_1$ from the pumping well). Test Case 5 is characterized by the presence of Type A wells in zones 1 and 2, and Type B wells in zone 3. Test Cases 6 and 7 are characterized by the presence of Type A wells in zone 1 and in zones 1-2 respectively, Type C wells being located in the remaining zones.

(2) *Group* 2. It includes 6 TCs (TC#2-TC#7) that are a variant of those of Group 1 and consider the solution of the data

assimilation procedure without neglecting flux exchanges between virtual monitoring boreholes of Type B and/or C and the aquifer, i.e., data assimilation is performed by considering perfect knowledge of the groundwater flow model, which includes all of the processes underpinning the reference head fields.

(3) *Group* 3. It includes 7 TCs designed to explore (*i*) the impacts of the mean and variance of the $Y$ reference field on the data assimilation results associated with Type B and C boreholes (TC2#[*]1-TC2#[*]2, TC3#[*]1-TC3#[*]2); and (*ii*) settings where

head data are assimilated solely from one depth (instead of all three locations) where a packer is installed along Type A wells (TC1[*]1-TC1[*]3).





In details, here we consider (*i*) a nearly uniform (while random) zero-mean *Y* reference field with variance equal to 0.01 (TC2#*1 and TC3#*1), the initial guesses employed for the variance and mean of *Y* during data assimilation being 0.09 and 0.2, respectively; (*ii*) a *Y* reference field with mean and variance equal to 0.2 and 1.70, respectively (TC2#*2 and TC3#*2),

the initial guesses employed for the variance and mean of *Y* during data assimilation being 1 and 0, respectively; and (*iii*) three variants of TC1: TC1*1 considers assimilating head information only from the upper packer (i.e., the one positioned at layer 4), TC1*2 and TC1*3 being designed to assimilate head data only from the intermediate (positioned at layer 7) and bottom (positioned at layer 13) packer, respectively.

     (4) *Group* 4. It includes 6 TCs where we explore the effect of inflating the measurement-error covariance matrix on the

data assimilation when the latter is performed in a way similar to the corresponding TC2 and TC3 of *Group* 1. As such, data assimilation is based on an imperfect flow model (where flux exchanges between the aquifer and monitoring boreholes are disregarded). To cope with this, inflation on measurement-error covariance matrix is considered during data assimilation, the inflation factor being set to $\alpha = 5$ (TC2$\alpha_1$ and TC3$\alpha_1$), 10 (TC2$\alpha_2$ and TC3$\alpha_2$), and 100 (TC2$\alpha_3$ and TC3$\alpha_3$; note that TC2 and TC3 correspond to $\alpha = 1$).

Initial input quantities required to solve moment equations and spatial fields of $K_G$ and $C_Y$ are obtained through the generation of 10,000 realizations of *Y*. The latter form the collection of realizations upon which the traditional Monte Carlo (MC)-based ensemble Kalman filter (MC-EnKF) is also applied. Results based on MEs-EnKF are then compared against those obtained through MC-EnKF. Head observations in all TCs are considered to be noisy and are obtained by adding a Gaussian white noise with a standard deviation of 0.01 to the reference heads collected at the virtual boreholes and used in

the data assimilation procedure. The strength of the noise is selected on the basis of the calculated reference head fields and considering the level of accuracy that is related to measuring devices commonly employed in practical settings (e.g., when considering water loggers, accuracy of pressure head observations is commonly comprised between $\sim \pm 0.005$ and $\sim \pm 0.05$ m).

     We rely on the criteria illustrated in the following to appraise the quality of the data assimilation performance. These are

(*i*) the average absolute difference between the estimated (or updated) *Y* field and its reference counterpart, $E_Y$, (*ii*) the square root of the average estimation variance, $S_Y$, and (*iii*) the average absolute difference between the estimated (or updated) aquifer head and its reference counterpart, $E_h$, evaluated as

$$E_Y = \frac{1}{N}\sum_{i=1}^{N}\left|\langle Y_i\rangle^u - Y_i^r\right| \tag{27}$$

$$S_Y = \sqrt{\frac{1}{N}\sum_{i=1}^{N}\left(\sigma_{Y,i}^2\right)^u} \tag{28}$$

$$E_h = \frac{1}{N}\sum_{i=1}^{N}\left|\langle h_i\rangle^u - h_i^r\right| \tag{29}$$





where $\langle Y_i \rangle^u$, $\left( \sigma_{Y,i}^2 \right)^u$ and $Y_i^r$ indicate the estimated mean, variance and reference $Y$ values at the $i^{th}$ node of the computational mesh, respectively; $\langle h_i \rangle^u$ and $h_i^r$ represent the estimated mean and reference aquifer head value at the $i^{th}$ node of the grid. We note that $S_Y$ is a metric quantifying the uncertainty associated with the estimated $Y$ field conditional on the data assimilated (see, e.g., Panzeri et al., 2014; and Nowak, 2010).

## 5 Results and discussion

### 5.1 Comparison between MC-EnKF and MEs-EnKF

In this Section we compare the results obtained with our MEs-EnKF approach and a standard MC-EnKF for two selected test cases, TC1 and TC2#. Table 2 summarize the outcomes computed via MEs-EnKF and MC-EnKF (increasing the number of MC simulations from 100 to 10,000) at the end of the assimilation process in terms of $E_Y$, $S_Y$, and $E_h$. These results suggest that the overall quality of conductivity estimates grounded on MEs-EnKF is similar to what one can obtain upon relying on a MC-EnKF based on 10,000 realizations, which is also consistent with the results illustrated by Panzeri et al. (2014) in two-dimensional settings. Table 2 also includes the computational cost (CPU in seconds) needed for each test case and approach using the processor Inter(R) Xeon(R) CPU E5-2650 v3 @ 2.30 GHz with 128 GB RAM. The CPU time required by MEs-EnKF is 20 times lower than the one required by a standard Monte Carlo based EnKF relying on 10,000 realization. When compared with the findings of Panzeri et al. (2014) in their two-dimensional settings, our results further support the computational appeal and feasibility of relying on a MEs-EnKF approach also in a three-dimensional setting. Note also that the CPU time required by TC2# is significantly larger (about six times) than the one needed for TC1, due to the implementation of flux exchanges between the aquifer system and the boreholes.

As an additional term of comparison, Fig. 4 depicts the spatial distributions of the estimated values of mean and variance the log-conductivity field computed with MEs-EnKF and MC- EnKF relying on 100, 500, 1,000 and 10,000 realizations at the end of the data assimilation window at layers 4, 7, and 10 (where the packers are located) in TC1. The reference $Y$ field is also depicted (see the left column of Fig. 4). Analogous outcomes are reported in Fig. 5 for TC2#. Visual inspection of these results provides further support to the ability of the MEs-EnKF to yield conductivity distributions consistent with the corresponding reference values in these settings. As expected, the degree of spatial variability of the estimated mean and variance values of $Y$ tends to stabilize as the number of realizations increases, being very similar to their MEs-based counterparts when 10,000 realizations are employed.



## 5.2 Effect of neglecting flux exchanges between boreholes and aquifer during data assimilation (*Group* 1)

Figure 6 shows the temporal behavior of $E_Y$ (Fig. 6a), $S_Y$ (Fig. 6b), and $E_h$ (Fig. 6c) for TCs1-7 (i.e., *Group* 1 in Table 1) obtained through MEs-EnKF. The lowest values of $E_Y$ are associated with TC1, where packers are set for all observation

wells. Very similar results are also obtained for TC5 and TC7, where Type B or C wells are installed only at the farthest locations (i.e., zone 3 in Fig. 3b) from the pumping well, respectively, Type A boreholes being installed within the regions (i.e., zones 1-2 in Fig. 3b) closest to the well.

The highest values of $E_Y$ correspond to TC3, where fully screened monitoring wells (Type C wells) are located in the entire domain. These are closely followed by the results associated with TC2, where observation wells screened across

multiple (Type B wells) levels are considered. Considering the trend displayed by the results in Fig. 6a, one can then conclude that relying on point values of head contributes to increase the overall quality of the data assimilation procedure, as expressed in terms of $E_Y$, when compared to considering vertically averaged head information while relying on a simplified groundwater flow model which neglects flux exchanges between screened intervals of boreholes and the surrounding aquifer.

Comparison between the values of $E_Y$ related to TC1 and TCs 5, 7 further suggests that the use of packers at locations far away (in terms of the horizontal correlation scale of $Y$) from the pumping well does not add additional information with respect to fully or partially screened wells, because vertical variability of head at such locations is modest. The importance of capturing vertical head variations during data assimilation is also manifest when comparing results related to TC4 and TC6. The values of $E_Y$ related to the former are consistently lower than those associated with the latter, a result which is

consistent with the use of fully screened (TC6) compared to partially screened (TC4) observation boreholes in most of the domain. The behavior of $E_h$ is very similar to the one displayed by $E_Y$, thus strengthening the above conclusions.

One can note that the scenarios characterized by a dominance of Type C boreholes (i.e., TC3 and TC6) are characterized by the lowest values of $S_Y$ (Fig. 6b). This result is related to the observation that depth-averaged well head information is here employed during data assimilation. Doing so tends to introduce a corresponding homogenization of the conductivity

field resulting from the data assimilation procedure, which is reflected by the lower values of $S_Y$. As such, while the results associated with $S_Y$ would suggest that the estimation variance associated with $Y$ is low, the overall accuracy, as given in terms of $E_Y$, is also low when relying mostly on vertically averaged data. Otherwise, temporal values of $S_Y$ are virtually indistinguishable for the other configurations considered.

The lowest $E_h$ values are visually indistinguishable and are related to TC1, TC5, and TC7, which is generally consistent

with the behavior of $E_Y$. The highest $E_h$ values are associated with TC3, where fully screened monitoring wells (Type C





wells) are located in the entire domain. These are followed by TC2, where partially screened monitoring wells (Type B wells) are distributed across the domain.

**5.3 Effect of including flux exchanges between boreholes and aquifer during data assimilation (*Groups* 2 and 3)**

Figure 7 depicts the temporal behavior of $E_Y$ (Fig. 7a), $S_Y$ (Fig. 7b), and $E_h$ (Fig. 7c) for TC1 and TC2#-TC7# (i.e., *Group*

2 in Table 1) obtained through MEs-EnKF. The lowest values of $E_Y$ are again associated with TC1. These are very closely matched by those obtained for TC5# and TC7#, where Type B or C wells are installed only at the farthest locations (i.e., zone 3 in Fig. 3b) from the pumping well. It is worth noting that the values of $E_Y$ in TCs 5# and 7# are virtually identical. The highest values of $E_Y$ correspond to TC3#, where only fully screened monitoring wells are distributed across the domain. These are very closely followed by the results associated with TC2#, where observation wells screened across multiple levels

(Type B wells) are considered. Note that the temporal evolution of $E_Y$ in these two TCs (TC2# and TC3#) is almost identical and tends to the same value at the end of the assimilation window. Comparison between values of $E_Y$ for TC4# and TC6# is also consistent with this finding.

Different from the results of *Group* 1, the temporal behavior of $S_Y$ (Fig. 7b) is very similar to one displayed by $E_Y$. These results are in line with (*i*) the observation that each Type A borehole provides three head observations, while only one well

head observation is essentially linked to Type B or C boreholes and (*ii*) the intuition that constraining the system with an increased number of observations would yield conductivity estimates characterized by an increased accuracy (in terms of lower $E_Y$ and possibly $S_Y$ values).

The lowest $E_h$ values are related to TC1, TC5# and TC7# and are visually indistinguishable (see Fig. 7c), a finding which is consistent with the behavior of $E_Y$ (see Fig. 7a). The highest $E_h$ values are associated with TC3#, closely followed by

TC2#, TC6#, and then TC4#. Otherwise, the values of $E_h$ become virtually indistinguishable.

Figure 7d depicts relative (percentage) differences between $E_Y$ evaluated for TCs2#-7# and TCs2-7, considering the values of TCs2-7 as references (negative values correspond to lower values of $E_Y$ in TCs2#-7# as compared with TCs2-7). Analogous results are reported for $S_Y$ (Fig. 7e) and $E_h$ (Fig. 7f). The largest accuracy improvement of $Y$ estimates, as suggested by Fig. 7d, corresponds to TC3# (where the inclusion of flux exchanges between boreholes and aquifer is

particularly relevant, since all monitoring wells fully penetrate the aquifer), followed by TC2#, TC6# and TC7#, with respect to their counterparts in *Group* 1.

Uncertainty associated with conductivity estimates increases in TCs2#-7# as compared against their counterparts in *Group* 1 (see Fig. 7e, where relative differences of $S_Y$ are all positive), this result being related to the vertical variability of flux





exchanges along Type B and C boreholes which is embedded in *Group* 2 TCs. The lowest (negative) relative differences for

$E_h$ correspond to TC3#, a result which is consistent with the depiction offered in Fig. 7d.

The temporal evolution of $E_Y$ (Fig. 8a), $S_Y$ (Fig. 8b), and $E_h$ (Fig. 8c) for TC2#*1, TC3#*1, TC2#*2 and TC3#*2 is displayed in Fig. 8. These results show that head observations collected at Type B or C screened wells yield conductivity and head estimates of similar quality (in terms of $E_Y$ and $S_Y$, or $E_h$, respectively) for the degrees of heterogeneity analyzed.

Figure 9 juxtaposes the temporal variability of $E_Y$ (Fig. 9a), $S_Y$ (Fig. 9b), and $E_h$ (Fig. 9c) values for TCs 1*1, 1*2 and

1*3 (see *Group* 3 in Table 1), TC1 (*Group* 1) and TC2# (*Group* 2). Values of $E_Y$ for TC2# are close to those of TC1*2 (where only data at the central layer of the system are assimilated). Otherwise, values of $E_Y$ at the end of the assimilation window for TC1*3 (where only data at the bottom of the system are assimilated) are lowest when considering the TCs TC1*1, TC1*2, TC1*3, and TC2#. Values of $S_Y$ for all test cases but TC1 are visually indistinguishable. Finally, the temporal behavior of $E_h$ for each test case is consistent with the one displayed by $E_Y$. These results seem to suggest that the

benefit (in terms of $E_Y$ and $E_h$) of collecting head observations from packers installed on multiple levels along the borehole depends on the duration of the assimilation period.

**5.4 Effect of inflation on measurement-error covariance matrix (*Group* 4)**

Figure 10 depicts the temporal evolution of $E_Y$ (Fig. 10a), $S_Y$ (Fig. 10b), and $E_h$ (Fig. 10c) for TC2$\alpha_1$, TC2$\alpha_2$ and TC2$\alpha_3$ (see *Group* 4 in Table 1), where partially screened monitoring wells (Type B wells) are located across the entire domain. The

lowest values of $E_Y$ are mainly associated with $\alpha = 10$ (i.e., TC2$\alpha_2$) while the highest values correspond to $\alpha = 1$ or 100 (i.e., TC2 or TC2$\alpha_3$, respectively). The magnitude of $S_Y$ is seen to increase with $\alpha$, in line with Eq. (24) according to which resorting to an inflation factor tends to decrease the strength of the dependence of conductivity estimates on head data. The highest $E_h$ values are generally linked to TC2 at observation times shorter than 10 (i.e., corresponding to stop of pumping), all other $E_h$ values being otherwise visually indistinguishable.

Based on these results, we conclude that the accuracy of conductivity and head estimates is generally improved when inflating the measurement-error covariance matrix. As stated above, these results are consistent with the observation that inflating the measurement-error covariance matrix results in a reduced weight of the mismatch between modeled and observed values during data assimilation. We recall that using inflation (i.e., setting $\alpha > 1$) may be useful to compensate for relying on an imperfect mathematical model, a scenario which is consistent with TC2-7 in *Group* 1. For instance, the

iterative ensemble smoothers (Chen et al., 2013; Luo et al., 2015) and the ensemble smoother with multiple data assimilation (ES-MDA; Emerick and Reynolds, 2013) rely on the action of an inflation factor on the measurement-error covariance matrix to cope with highly nonlinear systems.



Figure 11 shows the temporal behavior of $E_Y$ (Fig. 11a), $S_Y$ (Fig. 11b), and $E_h$ (Fig. 11c) for TC3 $\alpha_1$, TC3$\alpha_2$ and TC3$\alpha_3$ (see *Group* 4 in Table 1), where fully screened monitoring wells (Type C wells) are located in the entire domain. The lowest

$E_Y$ values are mainly associated with $\alpha = 100$ (i.e., TC3$\alpha_3$). The value of $E_Y$ at the end of the assimilation period is largest for TC3. The magnitude of $S_Y$ values tends to increase with $\alpha$ also in these cases. The highest and lowest $E_h$ values are generally linked to TC3 and TC3$\alpha_3$, respectively, for the early assimilation times and become virtually independent of $\alpha$ as time progresses. These results suggest that the highest inflation factors required to increase the quality of the data assimilation process (as measured through $E_Y$ and $E_h$) are associated with the scenarios where head data are collected in

fully screened boreholes (see, e.g., TC3$\alpha_3$ as compared against TC2$\alpha_2$).

**6 Conclusions**

We draw the following main conclusions based on this study:

● The use of packers to collect point-wise head data (Type A wells) yields higher accuracy of conductivity estimates than what can be obtained upon relying on partially or fully penetrating wells. The lowest values of $E_Y$ (average absolute

difference between the estimated mean of the logarithm of the conductivity field, $Y$, and its reference counterpart) are associated with the scenario where Type A wells are set across the domain.

● Using depth-averaged head data from partially (Type B wells) and fully screened (Type C wells) monitoring wells leads to comparable results in our settings, in terms of $E_Y$, $S_Y$ (square root of the average estimation variance of $Y$), and $E_h$ (average absolute difference between the estimated aquifer heads and their reference counterparts).

● Neglecting flux exchanges between the aquifer and partially/fully screened monitoring wells in the groundwater flow model can significantly deteriorate the accuracy of conductivity estimates. Considering the application of an inflation technique to measurement-error covariance matrix can improve conductivity estimates when an imperfect flow model is applied.

● The computational feasibility and accuracy of the moment equations-based ensemble Kalman filter (MEs-EnKF) are

explored. MEs-EnKF is as accurate as a typical Monte Carlo (MC) -based ensemble Kalman filter which relies on a large number (of the order of 10,000) of MC realizations. Otherwise, MEs-EnKF is more efficient than its MC-EnKF counterpart, the latter requiring about 20 times the central process unit (CPU) time of the former, on the basis of our examples.

**Acknowledgements**

This work was supported by the National Nature Science Foundation of China (Grant No. 41530316). Part of the work was developed while Prof. A. Guadagnini was at the University of Strasbourg with funding from Region Grand-Est and





Strasbourg-Eurometropole through the 'Chair Gutenberg'. Xiaodong Luo acknowledges financial support from the Research
Council of Norway through the Petromaks-2 project DIGIRES (RCN no. 280473) and the industrial partners AkerBP,
Wintershall DEA, Vår Energi, Petrobras, Equinor, Lundin and Neptune Energy. Chuan-An Xia was supported by
International Young Researcher Development Project of Guangdong Province, China.

**Appendix A: Cross-covariance between water levels in partially/fully screened monitoring boreholes**

The water level at well $I$, $h_I^w$ (with $I = 1, \ldots, N_w$), can be written as $h_I^w = \left\langle h_I^w \right\rangle + h_I'^w$. Making use of Eq. (10) one can obtain

the following expression for the water level fluctuation $h_I'^w$

$$h_I'^w \sum_{i=1}^{n} b_i K_i = \sum_{i=1}^{n} b_i K_i \left( h_i - \left\langle h_I^w \right\rangle \right) \tag{A1}$$

$n$ being the total number of cells according to which the screen of borehole $I$ is discretized. In a similar way, the water level

fluctuation at well $J$, $h_J'^w$, is given by

$$h_J'^w \sum_{i=j}^{m} b_j K_j = \sum_{j=1}^{m} b_j K_j \left( h_j - \left\langle h_J^w \right\rangle \right) \tag{A2}$$

where $m$ corresponds to the total number of cells according to which the screen of borehole $J$ is discretized. Multiplying Eq.
(A1) by Eq. (A2), taking expectation and disregarding moments of order larger than two yields

$$\left\langle h_I'^w h_J'^w \right\rangle \sum_{i=1}^{n} b_i \left\langle K_i \right\rangle \sum_{i=j}^{m} b_j \left\langle K_j \right\rangle =$$


$$= \sum_{j=1}^{m} \sum_{i=1}^{n} b_i b_j \left\{ \begin{array}{l} \left[ \left\langle K_i \right\rangle \left\langle K_j \right\rangle + \left\langle K_i' K_j' \right\rangle \right] \left[ \left\langle h_j \right\rangle \left\langle h_i \right\rangle - \left\langle h_J^w \right\rangle \left\langle h_i \right\rangle - \left\langle h_I^w \right\rangle \left\langle h_j \right\rangle + \left\langle h_I^w \right\rangle \left\langle h_J^w \right\rangle \right] \\ + \left[ \left\langle K_i \right\rangle \left\langle K_j' h_j' \right\rangle + \left\langle K_j \right\rangle \left\langle K_i' h_j' \right\rangle \right] \left\langle h_i \right\rangle \\ + \left[ \left\langle K_i \right\rangle \left\langle K_j' h_i' \right\rangle + \left\langle K_j \right\rangle \left\langle K_i' h_i' \right\rangle \right] \left\langle h_j \right\rangle \\ + \left[ \left\langle K_i \right\rangle \left\langle K_j \right\rangle \left\langle h_i' h_j' \right\rangle \right] - \left[ \left\langle K_i \right\rangle \left\langle K_j' h_i' \right\rangle + \left\langle K_j \right\rangle \left\langle K_i' h_{ii}' \right\rangle \right] \left\langle h_J^w \right\rangle \\ - \left[ \left\langle K_i \right\rangle \left\langle K_j' h_j' \right\rangle + \left\langle K_j \right\rangle \left\langle K_i' h_j' \right\rangle \right] \left\langle h_I^w \right\rangle \end{array} \right\} \tag{A3}$$

Evaluation of Eq. (A3) at second order yields Eq. (13).

*Code Availability*: The FORTRAN code used for solving moment equations of groundwater flow is available upon request.

*Author Contributions*: All authors make contribution to the preparation of the manuscript.

*Competing Interests*: The authors declare that they have no conflict of interest.



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






**Figures and Tables**

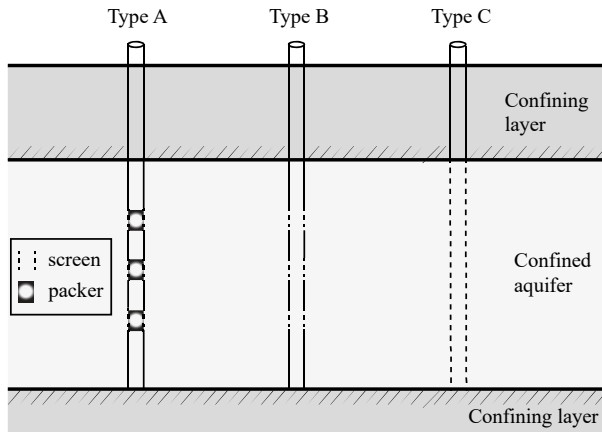

**Figure 1:** Type of monitoring wells: point-wise (Type A), partially (Type B) and fully penetrating (Type C) observation boreholes.


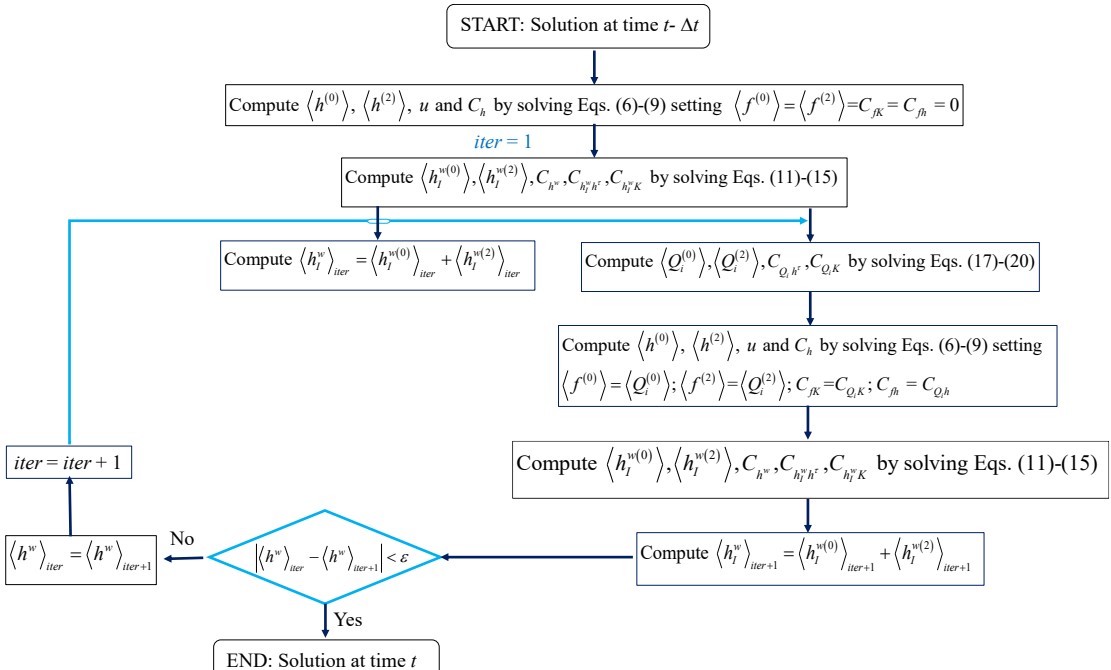

**Figure 2:** Workflow for the numerical solution of MEs within time interval [$t$-$\Delta t$, $t$] when flux between monitoring wells and the aquifer is considered.





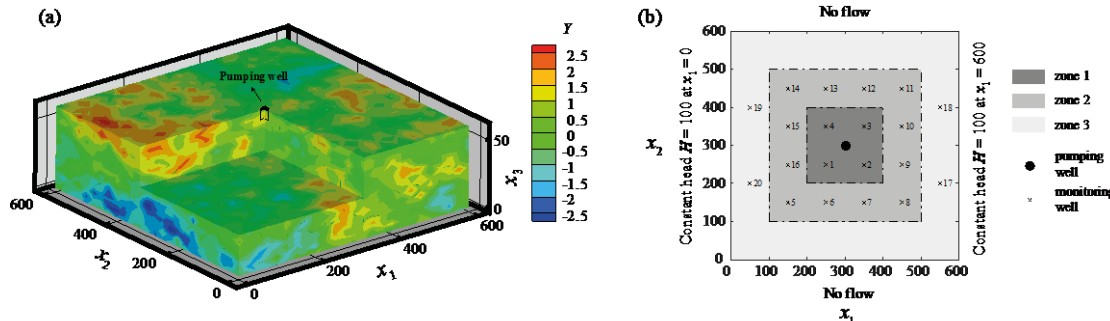


**Figure 3:** Reference log conductivity ($Y$) field (a) and location of monitoring wells within the flow domain (b).





**Figure 4:** Reference $Y$ field (left column) and estimates of mean and variance of $Y$ at the end of the assimilation process across layers 4, 7, and 10 for TC1. Results computed via MC-EnKF (with 100, 500, 1,000, and 10,000 realizations) and MEs-EnKF are included.





**Figure 5:** Reference $Y$ field (left column) and estimates of mean and variance of $Y$ at the end of the assimilation process at layers 4, 7, and 10 for TC2#. Results computed via MC-EnKF (with 100, 500, 1,000, and 10,000 realizations) and MEs-EnKF are included.






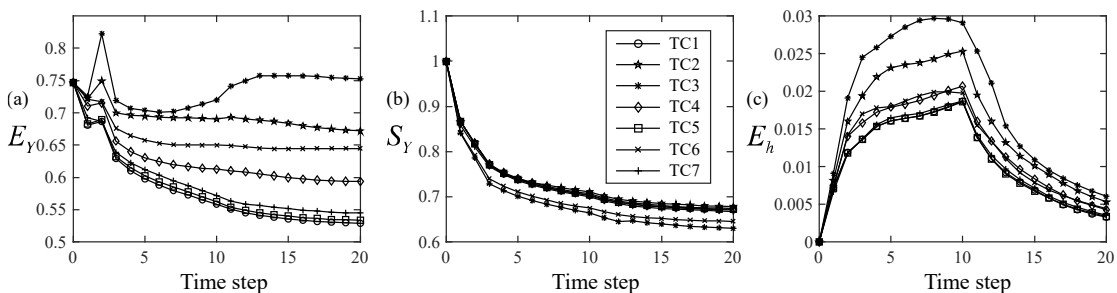

**Figure 6:** Temporal evolution of $E_Y$ (a), $S_Y$ (b), and $E_h$ (c) for TC1-TC7.

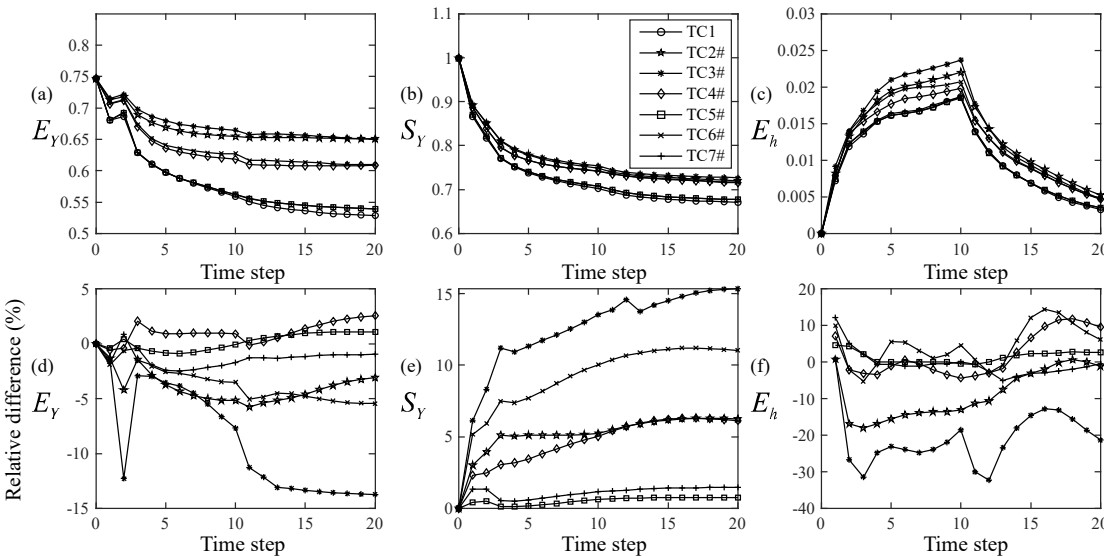

**Figure 7:** Temporal evolution of $E_Y$ (a), $S_Y$ (b), and $E_h$ (c) for TC1 and TC2#- TC7#. Temporal evolution of the relative difference

between $E_Y$ (d), $S_Y$ (e), and $E_h$ (f) evaluated in TC2#- TC7# and their counterparts related to TC2- TC7.





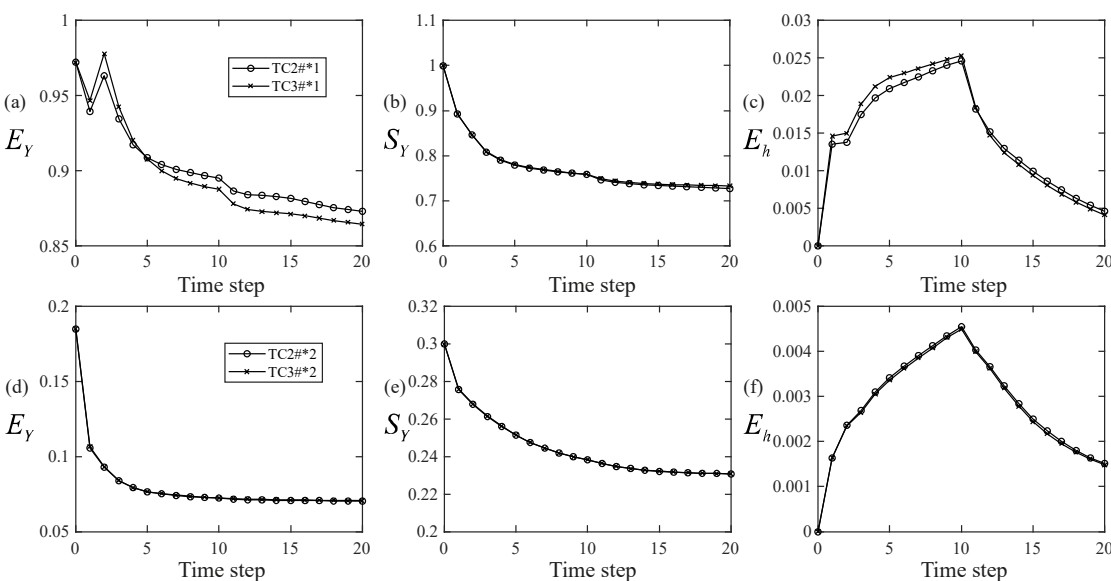

**Figure 8:** Temporal evolution of $E_Y$ (a, d), $S_Y$ (b, e), and $E_h$ (c, f) for TC2#*1 and TC3#*1 (top row) and for TC2#*2 and TC3#*2 (bottom row).

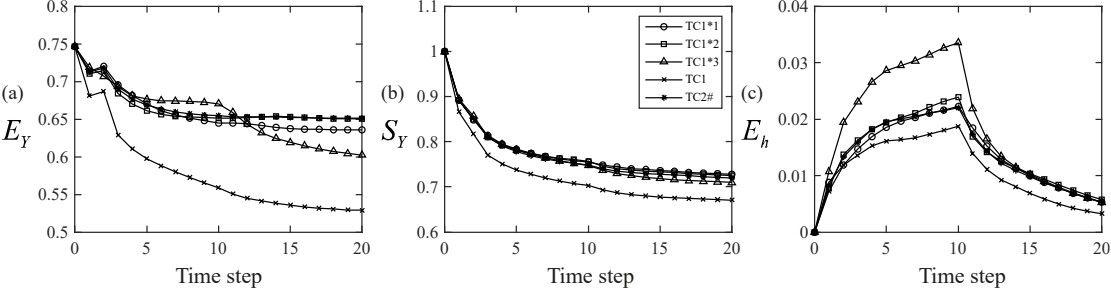

**Figure 9:** Temporal evolution of $E_Y$ (a), $S_Y$ (b), and $E_h$ (c) for TC1*1, TC1*2, TC1*3, TC1, and TC2#.

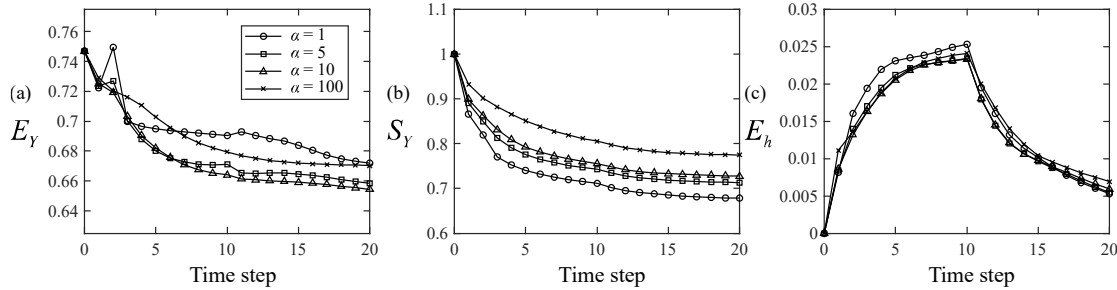

**Figure 10:** Temporal evolution of $E_Y$ (a), $S_Y$ (b), and $E_h$ (c) with $\alpha$ = 1 (TC2), 5 (TC2$\alpha_1$), 10 (TC2$\alpha_2$), and 100 (TC2$\alpha_3$).



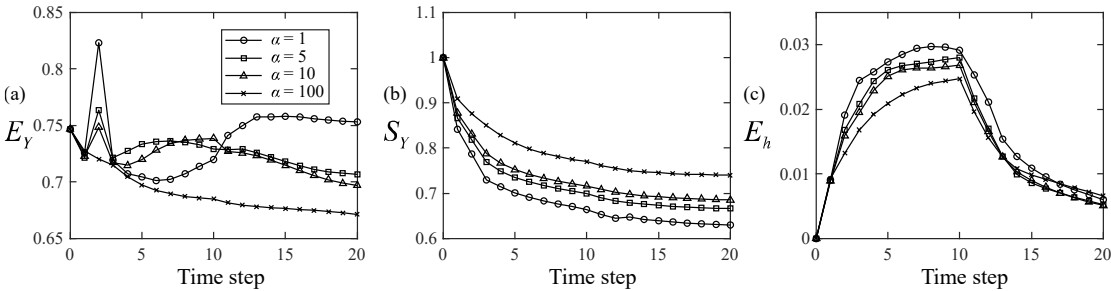

**Figure 11:** Temporal evolution of $E_Y$ (a), $S_Y$ (b), and $E_h$ (c) with $\alpha$ = 1 (TC3), 5 (TC3$\alpha_1$), 10 (TC3$\alpha_2$), and 100 (TC3$\alpha_3$).


**Table 1:** Summary of the Test Cases analyzed.

| TCs | | | | Type of monitoring wells | | |
|---|---|---|---|---|---|---|
| *Group* 1 | *Group* 2 | *Group* 3 | *Group* 4 | zone 1 | zone 2 | zone 3 |
| TC1 | | TC1*1, TC1*2, TC1*3 | | A | A | A |
| TC2 | TC2# | TC2#*1, TC2#*2 | TC2$\alpha_1$, TC2$\alpha_2$, TC2$\alpha_3$ | B | B | B |
| TC3 | TC3# | TC3#*1, TC3#*2 | TC3$\alpha_1$, TC3$\alpha_2$, TC3$\alpha_3$ | C | C | C |
| TC4 | TC4# | | | A | B | B |
| TC5 | TC5# | | | A | A | B |
| TC6 | TC6# | | | A | C | C |
| TC7 | TC7# | | | A | A | C |



**Table 2:** Comparison of results obtained by MEs-EnKF and MC-EnKF (based on 100, 500, 1,000, and 10,000 realizations) for TC1 and TC2#.

| TCs | Criterion | MC-EnKF | | | | MEs-EnKF |
|---|---|---|---|---|---|---|
| | | 100 | 500 | 1,000 | 10,000 | |
| TC1 | $E_Y$ | 0.84 | 0.59 | 0.55 | 0.53 | 0.53 |
| | $S_Y$ | 0.12 | 0.52 | 0.60 | 0.68 | 0.67 |
| | $E_h$ | 9.42E-3 | 5.60E-3 | 4.48E-3 | 3.75E-3 | 3.28E-3 |
| | CPU (sec) | 544 | 2,549 | 6,010 | 57,088 | 2,686 |
| TC2# | $E_Y$ | 0.80 | 0.69 | 0.66 | 0.65 | 0.65 |
| | $S_Y$ | 0.31 | 0.66 | 0.71 | 0.74 | 0.72 |
| | $E_h$ | 8.33E-3 | 5.91E-3 | 5.36E-3 | 5.30E-3 | 5.27E-3 |
| | CPU (sec) | 3,077 | 16,469 | 35,676 | 346,483 | 16,667 |