# Peer review of "Data assimilation with multiple types of observation boreholes via ensemble Kalman filter embedded within stochastic moment equations"

_Hydrology and Earth System Sciences, 2020_

## Referee Comment (RC1) · Anonymous Referee #1 · 20 Jan 2021

General Comments The paper refers to a numerical analysis, aimed at estimating the permeability field of a confined three-dimensional aquifer, based on head observations collected in piezometers that give different information. The aquifer conditions mimic the effects of a fully penetrating well pumping test that works for a certain time interval and the head recovery process following the ending of pumping. The monitoring piezometers considered are of three types: point detectors obtained with a multi packer device (type A), piezometers with multi-level sampling (type B) and fully penetrating wells for the entire thickness of the aquifer layers (type C). The reference aquifer is

heterogeneous with stationary second order pdf function (lognormal distribution) and exponential covariance. The variance of the field was assumed equal to 0.2 but some numerical tests were performed on a field with similar characteristics but with variance 1.70. The method used to solve the inverse problem is an Ensemble Kalman filter with stochastic moment equations (MEs-EnKF), already presented by the same authors in two-dimensional applications. A large number of numerical experiments were performed: a) varying the type of piezometers arranged in three areas of increasing distance from the well; b) differentiating the method of solving the stochastic equations to represent or not the flow exchange between the piezometers of type B and C and the surrounding aquifer; c) varying the variance of the field; d) evaluating the effect of an inflation coefficient. Finally, a comparison was made with the performance of a common Monte Carlo Ensemble Kalman Filter (MC-EnKF). The work aims to provide field operational indications such as the greater or lesser reliability of the types of piezometers investigated and on the methods of analysis through the evaluation of the results of the numerical tests performed.

Specific comments The work is strongly founded and explores topics of undoubted interest using consolidated methodologies which, in the current applications, are extended to three-dimensional cases. Some points for discussion can be the following. The proposed method MEs-EnKF is particularly convenient compared to the MC-EnKF with respect to the calculation times. However, being a perturbative method, albeit approximated to the second order, it presents the need for a limitation of the values of the variance. The tests carried out, aimed at evaluating the effect of high variances, explore the 0.2 - 1.7 range without going further. A point of interest is also the simulation of the flow between piezometers and the surrounding aquifer. It requires to set the dimensions of the effective radius and the radius of the well; it is not evident if the adopted dimensions should correspond to the real size of the borehole. Given that the piezometers with packers are more expensive and more complicated to install, one may wonder if the tests carried out suggest that it can be enough to install them only in zone 1 (closest to the pumping well) to obtain accetable reliability degree.

Technical correction It is necessary that the authors extend Table 1 by detailing the characteristics of the numerical experiment carried out for each test. This would be of great help to better interpret the graphs in figures 6-11 as well as the descriptive text in chapters 4 and 5. It is essential that Figures 6-11 are provided in color to better distinguish the different trends in the different tests. A detailed check of the correspondence between citations in the text and bibliographic references is necessary. The list of bibliographic references also needs accurate revision.
* * *

---

## Referee Comment (RC2) · Anonymous Referee #2 · 4 Feb 2021

General Comments

This research is orientated to a numerical flow modeling of a 3D confined aquifer. The scope of the exercise is to achieve the hydraulic conductivity field on a uniform flow system using different strategies for optimizing the analysis. The model is based on a tetrahedrons finite-element numerical solution with 13 layers. Some hydraulic parameters were imposed as constants as the variance of the hydraulic conductivity. In the domain a set of monitoring wells were arranged in order to give information about hydraulic heads. These wells were defined by three different types: 3-point sensors,

partly penetrating wells and fully penetrating boreholes.

The inverse problem was solved using two different methods: (i) Moment-Equations (MS) and (ii) Montecarlo Simulations (MC). Both methods were optimized via Ensemble Kalman Filter (EnKF). The exercise compares 4 different group of piezometers for 26 test cases analyzed taking into account different situations as: (i) neglecting flux exchanges, (ii) data achieved solely from a specific depth and (iii) the exploration of the effect of error in measurements.

As a result, a comparison on time-efficiency optimization method and the reliability on measurements of the implemented observation wells.

Specific comments

This is an interesting work based on a previous methodology implemented on 2D systems. This application shows us that MEs-EnKF has better time performance than MC-EnKF. Some assumptions were established as the size of the piezometers and the effective radius of the well. It is necessary to detail the units of each parameter and variable. It is also necessary to show the numerical features of each test. Finally, a formal review of citations and reference list is necessary. The figures that show the temporal evolution the parameters for appraising quality need would be done in color and bigger.

---

## Referee Comment (RC3) · Anonymous Referee #3 · 18 Feb 2021

General comments: The authors evaluate the accuracy of hydraulic conductivity (K) and head (h) estimates in a three dimensional, randomly heterogenous K field, when considering point (from multi-node monitoring wells) and depth averaged (from partially and fully screened monitoring wells) h measurements. The estimation of the K field is conducted via stochastic moment equations coupled with ensemble Kalman filter (ME-EnKF).

The authors first establish that, to solve this three-dimensional problem, the ME-EnKF approach is as accurate and computationally more efficient than EnKF relying on

10,000 Monte Carlo realizations/simulations. This result supports and extends previous findings from two-dimensional cases.

Then they use the ME-EnKF approach to investigate the importance of including point measurements in the assimilation process, leading to more accurate estimates of K and h fields, as opposed to employing depth averaged measurements. They also show that the accuracy of the results of the latter approach can be improved by using an inflation factor imposed to the observation error covariance matrix.

The manuscript is well written, logically structured and the conclusions are soundly supported by the results.

Specific comments: Second order approximations to moment equations are formally limited to sigma2_Y<1 or to well-conditioned, highly heterogenous media. Can the authors comment on their decision to place the observation wells at x-y distances close or equal to the value of the integral scale of Y?

Results for test cases in group 3 (sigma2_Y equal 0.2 and 1.7) are presented in Figure 8 but not discussed to the same level of detail than the rest of the cases. For example, it would be interesting to verify if the estimation errors in K and h increase with the variance of LnK (sigma2_Y).

Line 476, is "duration of the assimilation period" the appropriate term to refer to data collected at different depths (as in cases TC1*1, TC1*2, TC1*3)?

References cited in the text need to be checked (for example, line 86, Winter et al. (2003) is missing from the list of references, line 167, "Konikow . . . ").

---

## Author Comment (AC1) · 24 Feb 2021

Re: Revision of the manuscript " Data assimilation with multiple types of observation boreholes via ensemble Kalman filter embedded within stochastic moment equations" (Paper hess-2020-588) by Chuan-An Xia, Xiaodong Luo, Bill X. Hu, Monica Riva, Alberto Guadagnini.

Dear Referee#1:

We appreciate the efforts you have invested in our manuscript. Please, find in the

[Figure]

Supplement file an itemized list of your comments together with our response to each. Comments are listed in black font and our responses in blue font. Modifications implemented in the Revised Manuscript are indicated in red in the "Article Tracked Changes" document right after our responses.

Sincerely, Chuan-An Xia, Xiaodong Luo, Bill X. Hu, Monica Riva, Alberto Guadagnini

Please also note the supplement to this comment:
https://hess.copernicus.org/preprints/hess-2020-588/hess-2020-588-AC1-supplement.pdf

**Supplement:**

Feb, 24 2021

**Re: Revision of the manuscript " Data assimilation with multiple types of observation boreholes via ensemble Kalman filter embedded within stochastic moment equations" (Paper hess-2020-588) by Chuan-An Xia, Xiaodong Luo, Bill X. Hu, Monica Riva, Alberto Guadagnini.**

Dear Referee#1:

We appreciate the efforts you have invested in our manuscript. Please, find in the following an itemized list of your comments together with our response to each. Comments are listed in black font and our responses in blue font. Modifications implemented in the Revised Manuscript are indicated in red in the "Article Tracked Changes" document.

Sincerely,
Chuan-An Xia, Xiaodong Luo, Bill X. Hu, Monica Riva, Alberto Guadagnini
* * *
**General Comments**

The paper refers to a numerical analysis, aimed at estimating the permeability field of a confined three-dimensional aquifer, based on head observations collected in piezometers that give different information. The aquifer conditions mimic the effects of a fully penetrating well pumping test that works for a certain time interval and the head recovery process following the ending of pumping. The monitoring piezometers considered are of three types: point detectors obtained with a multi packer device (type A), piezometers with multi-level sampling (type B) and fully penetrating wells for the entire thickness of the aquifer layers (type C). The reference aquifer is heterogeneous with stationary second order pdf function (lognormal distribution) and exponential covariance. The variance of the field was assumed equal to 0.2 but some numerical tests were performed on a field with similar characteristics but with variance 1.70. The method used to solve the inverse problem is an Ensemble Kalman filter with stochastic moment equations (MEs-EnKF), already presented by the same authors in two-dimensional applications. A large number of numerical experiments were performed: a) varying the type of piezometers arranged in three areas of increasing distance from the well; b) differentiating the method of solving the stochastic equations to represent or not the flow exchange between the piezometers of type B and C and the surrounding aquifer; c) varying the variance of the field; d) evaluating the effect of an inflation coefficient. Finally, a comparison was made with the performance of a common Monte Carlo Ensemble Kalman Filter (MC-EnKF). The work aims to provide field operational indications such as the greater or lesser reliability of the types of piezometers investigated and on the methods of analysis through the evaluation of the results of the numerical tests performed.

    Answer: We truly appreciate your very insightful comments and positive evaluation.

**Specific comments**

The work is strongly founded and explores topics of undoubted interest using consolidated methodologies which, in the current applications, are extended to three-dimensional cases. Some points for discussion can be the following. The proposed method MEs-EnKF is particularly convenient compared to the MC-EnKF with respect to the calculation times.

    Answer: We agree with your points and appreciate your comments very much.

However, being a perturbative method, albeit approximated to the second order, it presents the need for a limitation of the values of the variance. The tests carried out, aimed at evaluating the effect of high variances, explore the 0.2 - 1.7 range without going further.

Answer: We further write that (lines 159-170) "It is worth noting that spatially heterogenous conductivities of aquifer systems are often modeled through a single, in some cases multimodal, distribution (Winter et al., 2003). This approach corresponds to a homogenization of conductivity values, which might be associated with diverse geomaterials, within a unique system. Otherwise, the domain can be conceptualized as composed by zones, each associated with a given geomaterial and hydrogeological attributes. This leads to modeling the system under investigation as composed by a collection of disjoint blocks, whose location might be uncertain and within which a quantity such as conductivity can be spatially heterogeneous (see e.g., Winter and Tartakovsky, 2000, 2002; Winter et al., 2002, 2003; Guadagnini et al., 2004; Short et al. 2010; Perulero Serrano et al., 2014; Bianchi Janetti et al., 2019 and references therein). In this framework one can represent conductivity within each block upon relying on a distribution associated with low to mild variance, which is compatible with the order of approximation associated with the groundwater flow moment equations we consider (Winter and Tartakovsky, 2002; Winter et al., 2002, 2003 and references therein). The scenario we investigate can then be seen as corresponding to the type of internal variability associated with a given geologic unit."

A point of interest is also the simulation of the flow between piezometers and the surrounding aquifer. It requires to set the dimensions of the effective radius and the radius of the well; it is not evident if the adopted dimensions should correspond to the real size of the borehole.

Answer: The concept of effective radius is related to the numerical scheme employed for the solution of the flow field, as seen, e.g., in the early work by Peaceman (1978) and in the most recent study by Chen and Zhang (2009), which we reference. We clarify this element by adding a corresponding reference. For the purpose of our analysis, we set the well radius $r_w$ = 0.1 (line 334), the corresponding value for the effective radius being $r_0$ = 2.81 (lines 332). These values are expressed in consistent units with all other quantities considered in the study.

While it is difficult to distinguish between the effects of the radius of the well casing and the effective radius of the area surrounding the casing (and including, e.g., grouting and/or gravel pack spaces) which is ascribable to well when in a numerical representation, our modeling choice roughly corresponds to average length scales associated with boreholes when lengths are provided, e.g., in meters. Given the ambiguity related to these concepts, we prefer to maintain the description of the set-up in consistent units.

With reference to the concept of well effective radius, our revised text now reads (lines 222-226) "Following Konikow et al. (2009), the link between $h_I^w$, $h_i$, and $Q_i$ can then be obtained by relying on the Thiem (1906) formulation as … where $r_0$ and $r_w$ are the effective (i.e., the radius of a well that would be associated with the same head as that calculated at the node of the cell that contains the well) and the actual well radius, respectively".

Given that the piezometers with packers are more expensive and more complicated to install, one may wonder if the tests carried out suggest that it can be enough to install them only in zone 1 (closest to the pumping well) to obtain acceptable reliability degree.

Answer: Our revised text now reads (lines 526-528 in our Conclusions) "It is additionally worth noting that the benefit of installing Type A wells as opposed to partially (Type B) or fully screened (Type C) monitoring wells is mainly associated with regions (e.g., zone 1 in this study) where strong variations of head along the vertical can take place.".

**Technical correction**
It is necessary that the authors extend Table 1 by detailing the characteristics of the numerical experiment carried out for each test. This would be of great help to better interpret the graphs in figures 6-11 as well as the descriptive text in chapters 4 and 5.

Answer: Prompted by the Reviewer's suggestion, our revised Table 1 now reads:

Table 1: Summary of the Test Cases analyzed.

| Groups | TCs | Type of monitoring well | | | Modeling approach for borehole/aquifer flux exchanges | Initial guess for log-conductivity fields | | Reference log-conductivity fields | | Inflation factor ($\alpha$) |
|---|---|---|---|---|---|---|---|---|---|---|
| | | zone 1 | zone 2 | zone 3 | | Mean | Variance | Mean | Variance | |
| *Group 1* | TC1 | A | A | A | Full model | 0.2 | 1.0 | 0.0 | 1.0 | 1.0 |
| | TC2 | B | B | B | Simplified model | 0.2 | 1.0 | 0.0 | 1.0 | 1.0 |
| | TC3 | C | C | C | Simplified model | 0.2 | 1.0 | 0.0 | 1.0 | 1.0 |
| | TC4 | A | B | B | Simplified model | 0.2 | 1.0 | 0.0 | 1.0 | 1.0 |
| | TC5 | A | A | B | Simplified model | 0.2 | 1.0 | 0.0 | 1.0 | 1.0 |
| | TC6 | A | C | C | Simplified model | 0.2 | 1.0 | 0.0 | 1.0 | 1.0 |
| | TC7 | A | A | C | Simplified model | 0.2 | 1.0 | 0.0 | 1.0 | 1.0 |
| *Group 2* | TC2# | B | B | B | Full model | 0.2 | 1.0 | 0.0 | 1.0 | 1.0 |
| | TC3# | C | C | C | Full model | 0.2 | 1.0 | 0.0 | 1.0 | 1.0 |
| | TC4# | A | B | B | Full model | 0.2 | 1.0 | 0.0 | 1.0 | 1.0 |
| | TC5# | A | A | B | Full model | 0.2 | 1.0 | 0.0 | 1.0 | 1.0 |
| | TC6# | A | C | C | Full model | 0.2 | 1.0 | 0.0 | 1.0 | 1.0 |
| | TC7# | A | A | C | Full model | 0.2 | 1.0 | 0.0 | 1.0 | 1.0 |
| *Group 3* | TC1*1 | A | A | A | Full model | 0.2 | 1.0 | 0.0 | 1.0 | 1.0 |
| | TC1*2 | A | A | A | Full model | 0.2 | 1.0 | 0.0 | 1.0 | 1.0 |
| | TC1*3 | A | A | A | Full model | 0.2 | 1.0 | 0.0 | 1.0 | 1.0 |
| | TC2#*1 | B | B | B | Full model | 0.2 | 0.09 | 0.0 | 0.01 | 1.0 |
| | TC2#*2 | B | B | B | Full model | 0.0 | 1.0 | 0.2 | 1.7 | 1.0 |
| | TC3#*1 | C | C | C | Full model | 0.2 | 0.09 | 0.0 | 0.01 | 1.0 |
| | TC3#*2 | C | C | C | Full model | 0.0 | 1.0 | 0.2 | 1.7 | 1.0 |
| *Group 4* | TC2$\alpha_1$ | B | B | B | Simplified model | 0.2 | 1.0 | 0.0 | 1.0 | 5.0 |
| | TC2$\alpha_2$ | B | B | B | Simplified model | 0.2 | 1.0 | 0.0 | 1.0 | 10.0 |
| | TC2$\alpha_3$ | B | B | B | Simplified model | 0.2 | 1.0 | 0.0 | 1.0 | 100.0 |
| | TC3$\alpha_1$ | C | C | C | Simplified model | 0.2 | 1.0 | 0.0 | 1.0 | 5.0 |
| | TC3$\alpha_2$ | C | C | C | Simplified model | 0.2 | 1.0 | 0.0 | 1.0 | 10.0 |
| | TC3$\alpha_3$ | C | C | C | Simplified model | 0.2 | 1.0 | 0.0 | 1.0 | 100.0 |

It is essential that Figures 6-11 are provided in color to better distinguish the different trends in the different tests.

Answer: We now plot Figs. 6-11 in color.

A detailed check of the correspondence between citations in the text and bibliographic references is necessary. The list of bibliographic references also needs accurate revision.

Answer: We have revised the correspondence between citations and reference list carefully. The reference list has been carefully checked and revised.

[revised manuscript text omitted]

For the purpose of our data assimilation workflow, we start by noting that we are interested in computing $C_h$ associated with two identical time coordinates, i.e., $C_h \left( \mathbf{y}, \mathbf{x}, \tau{=}t, t \right) = \left\langle h'(\mathbf{y}, \tau = t) h'(\mathbf{x}, t) \right\rangle^{(2)}$. We then recall that Zhang (2002) computes $C_h \left( \mathbf{y}, \mathbf{x}, \tau{=}t, t \right)$ for each time $t$ (while $C_h \left( \mathbf{y}, \mathbf{x}, \tau{=}t, t - \Delta t \right)$ is also unknown, $\Delta t$ being a constant temporal step size) by solving for $C_h \left( \mathbf{y}, \mathbf{x}, \tau{=}t, t' \right)$ from $t' = 0$ to $t' = t$. While this procedure can be computationally heavy for long times, Zhang (2002) points out that when flow changes only mildly, $C_h \left( \mathbf{x}, \mathbf{y}, \tau = t, t - \Delta t \right) \approx C_h \left( \mathbf{x}, \mathbf{y}, \tau = t - \Delta t, t - \Delta t \right)$, an approximation whose general validity is still not completely explored.

Here, we circumvent this issue and obtain high computational efficiency by directly evaluating $C_h \left( \mathbf{y}, \mathbf{x}, \tau{=}t, t \right)$ from $C_h \left( \mathbf{y}, \mathbf{x}, \tau{=}t - \Delta t, t - \Delta t \right)$ via (i) computing $C_h \left( \mathbf{y}, \mathbf{x}, \tau{=}t, t - \Delta t \right)$ through the solution of the equation obtained by considering Eq. (9) where the space and time derivatives operate on $\tau$ and $\mathbf{y}$ (instead of $t$ and $\mathbf{x}$) from time $t - \Delta t$ to $t$ using $C_h \left( \mathbf{y}, \mathbf{x}, \tau{=}t - \Delta t, t - \Delta t \right)$ as initial condition and then (ii) assessing $C_h \left( \mathbf{y}, \mathbf{x}, \tau{=}t, t \right)$ by solving Eq. (9) using $C_h \left( \mathbf{y}, \mathbf{x}, \tau{=}t, t - \Delta t \right)$ as initial condition.

It is further noted that Eqs. (6)-(9) are characterized by the same format, their discretization leading to a system of equations where the coefficients of the unknown quantities are identical, the corresponding right-hand-side terms (i.e., the forcing terms) being a function of the (ensemble) moment to be solved. In this context, one can resort to a direct solver for each time step. Thus, factorization of the matrix containing the coefficients of the system of equations is performed only

255    once, resulting in a high computational efficiency because only the right-hand-side term needs to be updated, depending on the moment of interest.

With reference to the forcing terms $\left\langle f^{(0)} \right\rangle$, $\left\langle f^{(2)} \right\rangle$, $C_{fK}$, and $C_{fh}$ in Eqs. (6)-(9), we note that these vanish for Type A wells and when one disregards flux exchanges between Type B (or C) wells and the aquifer. In these instances, mean head values and the associated covariance are simply obtained upon evaluating numerically Eqs. (6)-(9). Thus, when considering

260    a time interval $[t - \Delta t, t]$, the main computational cost stems from the evaluation of $u(\mathbf{y}, \mathbf{x}, t)$, $C_h(\mathbf{y}, \mathbf{x}, \tau = t, t - \Delta t)$, and $C_h(\mathbf{y}, \mathbf{x}, \tau = t, t)$, each of these requiring $N$ times the computational cost (hereafter denoted as $C_c^{\mathrm{MEs}}$) associated with the solution of the system of $N$ equations resulting after discretization. Therefore, the total computational effort required for solving Eqs. (6)-(9) at each time step is $3N C_c^{\mathrm{MEs}}$. Note that the computational effort is reduced to $2N C_c^{\mathrm{MEs}}$ for the first time interval, when the initial head is deterministic, or for a steady-state flow scenario (see Xia et al., 2019).

265    Otherwise, considering flux exchange processes when representing Type B (or C) wells entails evaluation of the source terms in Eqs. (6)-(9) as $\left\langle f^{(0)} \right\rangle = \left\langle Q_i^{(0)} \right\rangle$, $\left\langle f^{(2)} \right\rangle = \left\langle Q_i^{(2)} \right\rangle$, $C_{fk} = C_{Q_i K}$, and $C_{fh} = C_{Q_i h^\tau}$. The evaluation of the (ensemble) moments of interest across time interval $[t - \Delta t, t]$ is then performed through the workflow depicted in Fig. 2. In this case, we note that convergence of the iterative procedure is attained when the absolute difference between mean well heads at iteration $iter+1$, $\left\langle h_I^w \right\rangle_{iter+1}$, and $iter$, $\left\langle h_I^w \right\
[revised manuscript text omitted]

---

## Author Comment (AC2) · 24 Feb 2021

Re: Revision of the manuscript " Data assimilation with multiple types of observation boreholes via ensemble Kalman filter embedded within stochastic moment equations" (Paper hess-2020-588) by Chuan-An Xia, Xiaodong Luo, Bill X. Hu, Monica Riva, Alberto Guadagnini.

Dear Referee#2:

We appreciate the efforts you have invested in our manuscript. Please, find in the

Supplement file an itemized list of your comments together with our response to each. Comments are listed in black font and our responses in blue font. Modifications implemented in the Revised Manuscript are indicated in red in the "Article Tracked Changes" document right after our responses.

Sincerely, Chuan-An Xia, Xiaodong Luo, Bill X. Hu, Monica Riva, Alberto Guadagnini

Please also note the supplement to this comment:
https://hess.copernicus.org/preprints/hess-2020-588/hess-2020-588-AC2-supplement.pdf

[Figure]

**Supplement:**

Feb, 24 2021

**Re: Revision of the manuscript " Data assimilation with multiple types of observation boreholes via ensemble Kalman filter embedded within stochastic moment equations" (Paper hess-2020-588) by Chuan-An Xia, Xiaodong Luo, Bill X. Hu, Monica Riva, Alberto Guadagnini.**

Dear Referee#2:

We appreciate the efforts you have invested in our manuscript. Please, find in the following an itemized list of your comments together with our response to each. Comments are listed in black font and our responses in blue font. Modifications implemented in the Revised Manuscript are indicated in red in the "Article Tracked Changes" document.

Sincerely,
Chuan-An Xia, Xiaodong Luo, Bill X. Hu, Monica Riva, Alberto Guadagnini
* * *
**General Comments**
This research is orientated to a numerical flow modeling of a 3D confined aquifer. The scope of the exercise is to achieve the hydraulic conductivity field on a uniform flow system using different strategies for optimizing the analysis. The model is based on a tetrahedrons finite-element numerical solution with 13 layers. Some hydraulic parameters were imposed as constants as the variance of the hydraulic conductivity. In the domain a set of monitoring wells were arranged in order to give information about hydraulic heads. These wells were defined by three different types: 3-point sensors, partly penetrating wells and fully penetrating boreholes.

The inverse problem was solved using two different methods: (i) Moment-Equations (MS) and (ii) Montecarlo Simulations (MC). Both methods were optimized via Ensemble Kalman Filter (EnKF). The exercise compares 4 different group of piezometers for 26 test cases analyzed taking into account different situations as: (i) neglecting flux exchanges, (ii) data achieved solely from a specific depth and (iii) the exploration of the effect of error in measurements.

As a result, a comparison on time-efficiency optimization method and the reliability on measurements of the implemented observation wells.

Answer: We appreciate the assessment of the Reviewer.

**Specific comments**
This is an interesting work based on a previous methodology implemented on 2D systems. This application shows us that MEs-EnKF has better time performance than MC-EnKF. Some assumptions were established as the size of the piezometers and the effective radius of the well.

Answer: The concept of effective radius is related to the numerical scheme employed for the solution of the flow field, as seen, e.g., in the early work by Peaceman (1978) and in the most recent study by Chen and Zhang (2009), which we reference. We clarify this element by adding a corresponding reference. For the purpose of our analysis, we set the well radius $r_w$ = 0.1 (line 334), the corresponding value for the effective radius being $r_0$ = 2.81 (lines 332). These values are expressed in consistent units with all other quantities considered in the study. While it is difficult to distinguish between the effects of the radius of the well casing and the effective radius of the area surrounding the casing (and including, e.g., grouting and/or gravel pack spaces) which is ascribable to well when in a numerical representation, our modeling

choice roughly corresponds to average length scales associated with boreholes when lengths are provided, e.g., in meters. Given the ambiguity related to these concepts, we prefer to maintain the description of the set-up in consistent units.

With reference to the concept of well effective radius, our revised text now reads (lines 222-226) "Following Konikow et al. (2009), the link between $h_I^w$, $h_i$, and $Q_i$ can then be obtained by relying on the Thiem (1906) formulation as … where $r_0$ and $r_w$ are the effective (i.e., the radius of a well that would be associated with the same head as that calculated at the node of the cell that contains the well) and the actual well radius, respectively".

References
Konikow, L. F., Hornberger, G. Z., Halford, K. J., Hanson, R. T., and Harbaugh, A. W.: Revised multi-node well (MNW2) package for MODFLOW ground-water flow model, Report 6-A30, 2009.
Peaceman, D.W.: Interpretation of well-block pressures in numerical reservoir simulation, Soc. Pet. Eng. J., 18(3), 319-324.
Thiem, G.: Hydrologische methoden, Leipzig, Germany, J.M. Gebhart, 56 p, 1906.

It is necessary to detail the units of each parameter and variable.

Answer: Consistent with prior studies, which we reference in the manuscript (e.g., Zhang, 2002; Li and Tchelepi, 2006; Panzeri et al., 2013 and references therein), all quantities are intended to be in consistent space-time units, which are therefore omitted. We explicitly address this issue in our revised text which now reads (lines 310-312) "We consider a three-dimensional domain (Fig. 3a) of size $600\times600\times60$ (hereafter, all quantities are considered in consistent units, following notation associated with prior studies, including, e.g., Panzeri et al., 2013 and references therein), the system being discretized onto a numerical mesh comprising $25\times25\times13$ nodes, for a total of 34,560 tetrahedrons."

It is also necessary to show the numerical features of each test.

Answer: Prompted by the Reviewer's suggestion, our revised Table 1 now reads:

**Table 1:** Summary of the Test Cases analyzed.

| Groups | TCs | Type of monitoring well | | | Modeling approach for borehole/aquifer flux exchanges | Initial guess for log-conductivity fields | | Reference log-conductivity fields | | Inflation factor ($\alpha$) |
|---|---|---|---|---|---|---|---|---|---|---|
| | | zone 1 | zone 2 | zone 3 | | Mean | Variance | Mean | Variance | |
| Group 1 | TC1 | A | A | A | Full model | 0.2 | 1.0 | 0.0 | 1.0 | 1.0 |
| | TC2 | B | B | B | Simplified model | 0.2 | 1.0 | 0.0 | 1.0 | 1.0 |
| | TC3 | C | C | C | Simplified model | 0.2 | 1.0 | 0.0 | 1.0 | 1.0 |
| | TC4 | A | B | B | Simplified model | 0.2 | 1.0 | 0.0 | 1.0 | 1.0 |
| | TC5 | A | A | B | Simplified model | 0.2 | 1.0 | 0.0 | 1.0 | 1.0 |
| | TC6 | A | C | C | Simplified model | 0.2 | 1.0 | 0.0 | 1.0 | 1.0 |
| | TC7 | A | A | C | Simplified model | 0.2 | 1.0 | 0.0 | 1.0 | 1.0 |
| Group 2 | TC2# | B | B | B | Full model | 0.2 | 1.0 | 0.0 | 1.0 | 1.0 |
| | TC3# | C | C | C | Full model | 0.2 | 1.0 | 0.0 | 1.0 | 1.0 |
| | TC4# | A | B | B | Full model | 0.2 | 1.0 | 0.0 | 1.0 | 1.0 |
| | TC5# | A | A | B | Full model | 0.2 | 1.0 | 0.0 | 1.0 | 1.0 |
| | TC6# | A | C | C | Full model | 0.2 | 1.0 | 0.0 | 1.0 | 1.0 |
| | TC7# | A | A | C | Full model | 0.2 | 1.0 | 0.0 | 1.0 | 1.0 |
| Group 3 | TC1*1 | A | A | A | Full model | 0.2 | 1.0 | 0.0 | 1.0 | 1.0 |
| | TC1*2 | A | A | A | Full model | 0.2 | 1.0 | 0.0 | 1.0 | 1.0 |
| | TC1*3 | A | A | A | Full model | 0.2 | 1.0 | 0.0 | 1.0 | 1.0 |
| | TC2#*1 | B | B | B | Full model | 0.2 | 0.09 | 0.0 | 0.01 | 1.0 |
| | TC2#*2 | B | B | B | Full model | 0.0 | 1.0 | 0.2 | 1.7 | 1.0 |
| | TC3#*1 | C | C | C | Full model | 0.2 | 0.09 | 0.0 | 0.01 | 1.0 |
| | TC3#*2 | C | C | C | Full model | 0.0 | 1.0 | 0.2 | 1.7 | 1.0 |
| Group 4 | TC2$\alpha_1$ | B | B | B | Simplified model | 0.2 | 1.0 | 0.0 | 1.0 | 5.0 |
| | TC2$\alpha_2$ | B | B | B | Simplified model | 0.2 | 1.0 | 0.0 | 1.0 | 10.0 |
| | TC2$\alpha_3$ | B | B | B | Simplified model | 0.2 | 1.0 | 0.0 | 1.0 | 100.0 |
| | TC3$\alpha_1$ | C | C | C | Simplified model | 0.2 | 1.0 | 0.0 | 1.0 | 5.0 |
| | TC3$\alpha_2$ | C | C | C | Simplified model | 0.2 | 1.0 | 0.0 | 1.0 | 10.0 |
| | TC3$\alpha_3$ | C | C | C | Simplified model | 0.2 | 1.0 | 0.0 | 1.0 | 100.0 |

Finally, a formal review of citations and reference list is necessary.

Answer: Many thanks. We have rechecked the correspondence between citation and reference list. The related revisions are tracked in the document of track change.

The figures that show the temporal evolution the parameters for appraising quality need would be done in color and bigger.

Answer: We have replotted Figs. 6-11 (see the revised document) in color with larger size in comparison to their original versions.

[revised manuscript text omitted]
{\langle K_i' h_I'^w \rangle^{(2)}}{K_{G,i}} + \langle h_i^{(2)} \rangle + \frac{\sigma_{Y,i}^2}{2} \left( \langle h_i^{(0)} \rangle - \langle h_I^{w(0)} \rangle \right) \right\} \tag{12}$$

Here, $T_{G,i} = b_i K_{G,i}$, $\langle h_i^{(0)} \rangle$, $\langle h_i^{(2)} \rangle$, $K_{G,i}$ and $\sigma_{Y,i}^2$ correspond to the zero- (evaluated by Eq. (6)) and the second- (evaluated by Eq. (7)) order mean head, geometric mean of conductivity and variance of log-conductivity at the $i^{th}$ cell of a multi-node monitoring well, respectively; $u_{ii} = \langle K_i' h_i' \rangle^{(2)}$ is the cross-covariance between conductivity and head at the $i^{th}$ cell; $\langle K_i' h_I'^w \rangle^{(2)}$ is the cross-covariance between well head and conductivity at the $i^{th}$ cell (evaluated as detailed below, see Eq. (15)).

The covariance between water levels at wells $I$ (i.e., $h_I^w$) and $J$ (i.e., $h_J^w$), $C_{h^w} = \langle h_I'^w h_J'^w \rangle^{(2)}$, can be evaluated as (see also Appendix A, Eqs. (A1)-(A3))

$$C_{h^w} \sum_{i=1}^{n} T_{G,i} \sum_{j=i}^{m} T_{G,j} =$$

$$= \sum_{j=1}^{m} \sum_{i=1}^{n} T_{G,i} T_{G,j} \left\{ \begin{array}{l} \left( \langle h_i^{(0)} \rangle - \langle h_I^{w(0)} \rangle \right) \left( \langle h_j^{(2)} \rangle - \langle h_J^{w(2)} \rangle + \dfrac{u_{jj}}{K_{G,j}} + \dfrac{u_{ij}}{K_{G,i}} \right) \\[3mm] + \left( \langle h_j^{(0)} \rangle - \langle h_J^{w(0)} \rangle \right) \left( \langle h_i^{(2)} \rangle - \langle h_I^{w(2)} \rangle + \dfrac{u_{ji}}{K_{G,j}} + \dfrac{u_{ii}}{K_{G,i}} \right) \\[3mm] + \left( \dfrac{\sigma_{Y,i}^2}{2} + \dfrac{\
[revised manuscript text omitted]
}\rangle \sum_{i=1}^{n} b_i \langle K_i\rangle \sum_{i=j}^{m} b_j \langle K_j\rangle =$$

$$= \sum_{j=1}^{m}\ \sum_{i=1}^{n} b_i b_j \left\{ \begin{array}{l} \left[\langle K_i\rangle\langle K_j\rangle + \langle K_i' K_j'\rangle\right]\left[\langle h_j\rangle\langle h_i\rangle - \langle h_J^w\rangle\langle h_i\rangle - \langle h_I^w\rangle\langle h_j\rangle + \langle h_I^w\rangle\langle h_J^w\rangle\right] \\[4pt] + \left[\langle K_i\rangle\langle K_j' h_j'\rangle + \langle K_j\rangle\langle K_i' h_j'\rangle\right]\langle h_i\rangle \\[4pt] + \left[\langle K_i\rangle\langle K_j' h_i'\rangle + \langle K_j\rangle\langle K_i' h_i'\rangle\right]\langle h_j\rangle \\[4pt] + \left[\langle K_i\rangle\langle K_j\rangle\langle h_i' h_j'\rangle\right] - \left[\langle K_i\rangle\langle K_j' h_i'\rangle + \langle K_j\rangle\langle K_i' h_{ii}'\rangle\right]\langle h_J^w\rangle \\[4pt] - \left[\langle K_i\rangle\langle K_j' h_j'\rangle + \langle K_j\rangle\langle K_i' h_j'\rangle\right]\langle h_I^w\
[revised manuscript text omitted]

---

## Author Comment (AC3) · 24 Feb 2021

Re: Revision of the manuscript " Data assimilation with multiple types of observation boreholes via ensemble Kalman filter embedded within stochastic moment equations" (Paper hess-2020-588) by Chuan-An Xia, Xiaodong Luo, Bill X. Hu, Monica Riva, Alberto Guadagnini.

Dear Referee#3:

We appreciate the efforts you have invested in our manuscript. Please, find in the

[Figure]

Supplement file an itemized list of your comments together with our response to each. Comments are listed in black font and our responses in blue font. Modifications implemented in the Revised Manuscript are indicated in red in the "Article Tracked Changes" document right after our responses.

Sincerely, Chuan-An Xia, Xiaodong Luo, Bill X. Hu, Monica Riva, Alberto Guadagnini

Please also note the supplement to this comment:
https://hess.copernicus.org/preprints/hess-2020-588/hess-2020-588-AC3-supplement.pdf

**Supplement:**

Feb, 24 2021

**Re: Revision of the manuscript " Data assimilation with multiple types of observation boreholes via ensemble Kalman filter embedded within stochastic moment equations" (Paper hess-2020-588) by Chuan-An Xia, Xiaodong Luo, Bill X. Hu, Monica Riva, Alberto Guadagnini.**

Dear Referee#3:

We appreciate the efforts you have invested in our manuscript. Please, find in the following an itemized list of your comments together with our response to each. Comments are listed in black font and our responses in blue font. Modifications implemented in the Revised Manuscript are indicated in red in the "Article Tracked Changes" document.

Sincerely,
Chuan-An Xia, Xiaodong Luo, Bill X. Hu, Monica Riva, Alberto Guadagnini
* * *
**General Comments**
The authors evaluate the accuracy of hydraulic conductivity (K) and head (h) estimates in a three dimensional, randomly heterogenous K field, when considering point (from multi-node monitoring wells) and depth averaged (from partially and fully screened monitoring wells) h measurements. The estimation of the K field is conducted via stochastic moment equations coupled with ensemble Kalman filter (MEEnKF). The authors first establish that, to solve this three-dimensional problem, the ME-EnKF approach is as accurate and computationally more efficient than EnKF relying on 10,000 Monte Carlo realizations/simulations. This result supports and extends previous findings from two-dimensional cases. Then they use the ME-EnKF approach to investigate the importance of including point measurements in the assimilation process, leading to more accurate estimates of K and h fields, as opposed to employing depth averaged measurements. They also show that the accuracy of the results of the latter approach can be improved by using an inflation factor imposed to the observation error covariance matrix. The manuscript is well written, logically structured and the conclusions are soundly supported by the results.

> Answer: We truly appreciate the very positive evaluation.

**Specific comments**
Second order approximations to moment equations are formally limited to sigma2_Y<1 or to well-conditioned, highly heterogenous media. Can the authors comment on their decision to place the observation wells at x-y distances close or equal to the value of the integral scale of Y?

> Answer: Expected values and covariances of heads (hence drawdowns) driven by pumping are governed by conductivity correlation (integral) scale. Examples associated with analytical solutions of steady-state two- and three-dimensional convergent flow based on Moment Equations and supporting these aspects are given by Riva et al. (2001) and Guadagnini et al. (2003). As such, we opted to distribute observation boreholes at distances from the pumping well approximately corresponding to multiples of the log-conductivity correlation scale.
> Our revised text now reads (lines 355-357): "We note that the spatial arrangement of the observation boreholes is designed to allow these to be spaced by a distance approximately corresponding to a correlation scale of *Y*, thus encompassing strong to low degrees of correlation with respect to the pumping well location."

Results for test cases in group 3 (sigma2_Y equal 0.2 and 1.7) are presented in Figure 8 but not discussed to the same level of detail than the rest of the cases. For example, it would be interesting to verify if the estimation errors in K and h increase with the variance of LnK (sigma2_Y).

Answer: Prompted by the Reviewer's comments, we have enhanced the description of the results included in Figure 8. The revised text now reads (lines 483-488) "Mean absolute differences between $E_Y$ values associated with TC2#*1 and TC3#*1 is 0.008, while being virtually null when considering TC2#*2 and TC3#*2. Results of corresponding quality are also obtained when comparing $S_Y$ and $E_h$ values related to TC2#*1 and TC3#*1, or TC2#*2 and TC3#*2. We further note that values of $E_Y$ in Fig. 8d (or $E_h$ in Fig. 8f) are always higher than their counterparts depicted in Fig. 8a (or Fig. 8c), consistent with the observation that the accuracy of conductivity (and head) estimates tends to deteriorate with increasing degree of spatial heterogeneity of conductivities.

Line 476, is "duration of the assimilation period" the appropriate term to refer to data collected at different depths (as in cases TC1*1, TC1*2, TC1*3)?

Answer: We now write (lines 494-496) "These results seem to suggest that the benefit (in terms of $E_Y$ and $E_h$) of collecting head observations from packers installed along the borehole depends on the observation depth and on the duration of the assimilation period.".

References cited in the text need to be checked (for example, line 86, Winter et al. (2003) is missing from the list of references, line 167, "Konikow : : : ").

Answer: We have revised the correspondence between citations and reference list carefully. The reference list has been carefully checked and revised.

[revised manuscript text omitted]